# Extracellular DNA released by glycine-auxotrophic *Staphylococcus epidermidis* small colony variant facilitates catheter-related infections

Junlan Liu[1], Zhen Shen[1], Jin Tang[2], Qian Huang[1], Ying Jian[1], Yao Liu[1], Yanan Wang[1], Xiaowei Ma[1], Qian Liu[1], Lei He [1]✉ & Min Li [1,3]✉

Though a definitive link between small colony variants (SCVs) and implant-related staphylococcal infections has been well-established, the specific underlying mechanism remains an ill-explored field. The present study analyzes the role SCVs play in catheter infection by performing genomic and metabolic analyses, as well as analyzing biofilm formation and impacts of glycine on growth and peptidoglycan-linking rate, on a clinically typical *Staphylococcus epidermidis* case harboring stable SCV, normal counterpart (NC) and nonstable SCV. Our findings reveal that *S. epidermidis* stable SCV carries mutations involved in various metabolic processes. Metabolome analyses demonstrate that two biosynthetic pathways are apparently disturbed in SCV. One is glycine biosynthesis, which contributes to remarkable glycine shortage, and supplementation of glycine restores growth and peptidoglycan-linking rate of SCV. The other is overflow of pyruvic acid and acetyl-CoA, leading to excessive acetate. SCV demonstrates higher biofilm-forming ability due to rapid autolysis and subsequent eDNA release. Despite a remarkable decline in cell viability, SCV can facilitate in vitro biofilm formation and in vivo survival of NC when co-infected with its normal counterparts. This work illustrates an intriguing strategy utilized by a glycine-auxotrophic clinical *S. epidermidis* SCV isolate to facilitate biofilm-related infections, and casts a new light on the role of SCV in persistent infections.

[1] Department of Laboratory Medicine, Renji Hospital, School of Medicine, Shanghai Jiao Tong University, Shanghai 200127, China. [2] Department of Laboratory Medicine, Shanghai Jiao Tong University Affiliated Sixth People's Hospital, Shanghai 200233, China. [3] Faculty of Medical Laboratory Science, Shanghai Jiao Tong University School of Medicine, Shanghai 200025, China. ✉email: buningweishi_1985@126.com; rjlimin@shsmu.edu.cn

Small colony variants (SCVs) represent a slowly growing subpopulation of bacteria that has been recovered from an extensive range of species for many decades[1–5]. Nevertheless, not until recent decades did researchers begin to realize the key role that SCVs play in persistent infections[6–8]. SCVs were thereafter tightly linked to persistence and relapse of all manner of infections[8–13]. In the context of biofilm-related catheter infections, however, the underlying strategy utilized by SCVs to facilitate occurrence and persistence of infection has thus far remained ill-explored and enigmatic. Given that infections caused by catheters and many other types of indwelling medical devices have long been recognized as predominant origins of life-threatening nosocomial sepsis[14,15], it is increasingly urgent to elucidate the contribution of SCV to occurrence and development of catheter-related infections.

By consequence of its transcendent biofilm-forming capability, the skin commensal *Staphylococcus epidermidis* has become the leading etiologic agent not just of catheter infections, but more importantly of deadly catheter-related bloodstream infections[16–18]. Biofilm formation requires an array of activities including adherence to catheter surfaces and subsequent aggregation into mature structure, wherein diverse matrix molecules play an essential part[16]. The three dimensional, multicellular, and multilayered biofilm formed by *S. epidermidis* shields it from external detrimental agents, thereby allowing *S. epidermidis* to persist within the host even under exposure to active antibiotics and efficient immune defense[16,19]. With the widespread application of catheterization in hospitals around the world, infection caused by *S. epidermidis* is becoming more of an issue than ever before[20,21].

There is increasing evidence demonstrating that *S. epidermidis* SCVs from clinical medical device-related infections evolve from a normal-sized clone during the course of infection, and exhibits higher adaptiveness than the parental clone[4,11,22–24]. Further, a definitive link between SCVs and persistence of implant-related staphylococcal infections has been well established[25,26]. However, the existing understanding of *S. epidermidis* SCVs remains limited in scope, especially within genetically undefined clinical *S. epidermidis* SCV isolates. Since mutations in genes involved in the electron transport chain (ETC), such as *hemB* and *menD*, have been implicated in the formation of SCVs, corresponding mutants have been recreated for characterization of SCVs[27,28]. Although these stable SCVs mutants have provided profound insights into lifestyle of SCVs, of particular note is that mutations associated with SCVs formation are usually unstable[29,30]. As such, SCVs can easily revert to normal colony phenotype via reversal of mutations or gain of second-site mutations[30,31]. In other words, most SCVs are phenotypically unstable after serial passages. Obviously, such instability hampers in-depth characterization of SCVs, thereby leading to the fact that numerous studies established their conclusions based on phenotypically stable ETC mutants[27,32]. Still, with the advent of the omics era, the burgeoning omics technologies have provided a wider scope of information about SCVs, highlighting the fact that the proteomic and mutational profiles of clinical SCVs are both distinct from and more complicated than those of genetically defined ETC mutants[33,34]. Such diversity in their phenotypes further underscores the significance of clinical SCVs based investigation, which is more clinically relevant and translatable to the management of persistent infections associated with SCVs.

Metabolic dysfunction and SCVs have always been inextricably intertwined[35,36]. In addition to ETC, various metabolic pathways, such as those involved in lipid biosynthesis, are also able to result in growth arrest, either directly or indirectly[37]. But as to other phenotypes apart from growth defect, downregulation of the quorum-sensing system Agr has consistently been considered as the major cause[35]. Nonetheless, inexplicable phenotypes of clinical SCVs still persist[22,38], indicating that the currently known mechanisms underlying the pathogenicity of SCVs are far from being comprehensive.

Thus, the present work selected a clinically typical *S. epidermidis* SCV case of central venous catheter infection, which is rarely defined but significant in biofilm-related infections, as the subject to probe into its potential role in catheter infections. On the strength of the combination of genomics and metabolomics approaches, we discover that this clinical *S. epidermidis* SCV isolate demonstrates a seldom-reported glycine auxotrophy. The present study also reveals an intriguing strategy that *S. epidermidis* SCV utilized in promoting the development of biofilm-related catheter infections.

## Results

**High prevalence of SCVs among *S. epidermidis*-dominated catheter infections.** A total of 128 cases of catheter infection were collected from multiple medical centers in Shanghai over a half-year period. Staphylococci, especially *S. epidermidis*, accounted for 39.1% of total cases, whereas gram-negative bacilli, corynebacteria, and enterococci only represented 18.0%, 4.7%, and 3.9%, respectively, of cases (Fig. 1a). Notably, 21.9% of catheter infections were due to multispecies, moreover, the most common coinfection pattern was *S. epidermidis* related (14/28, 50%), which further demonstrates the dominant role of *S. epidermidis* in catheter infections (Fig. 1a).

Of particular interest, SCVs frequently accompanied its normal counterparts (NCs) in the same case, with an extremely high prevalence (41/50, 82%) among staphylococcal infections (Fig. 1b) (Table 1). Nevertheless, the majority of SCVs from initial catheter cultures were phenotypically unstable SCVs, as they could easily revert to normal-sized colonies after being subcultured, which is also referred to as revertant (Rev) (Fig. 1b). Accordingly, only 8 out of 50 cases had stable SCVs after stability check (Table 1).

**Enhanced biofilm formation of catheter-sourced clinical *S. epidermidis* stable SCV.** Besides colony morphological variation, other phenotypic characteristics, including increased antibiotic resistance and altered pathogenicity, are also typical features of clinical SCVs[4,11,22,39]. Yet undoubtedly, the instability of most clinical SCVs from catheters poses a great barrier to acquiring further insights into the role of *S. epidermidis* SCVs in catheter infections. Therefore, a catheter-sourced stable SCV isolate of *S. epidermidis* was selected for subsequent analyses, together with its NC and a nonstable SCV isolate (Rev) from one and the same patient (Supplementary Table 1). This *S. epidermidis* stable SCV isolate was selected for the reason that it represents the predominant clone in clinical *S. epidermidis* (i.e., MRSE-ST2)[40,41].

Interestingly, the *S. epidermidis* stable SCV isolate did not demonstrate such frequently reported auxotrophy as hemin, thymidine, and menadione auxotrophy (Supplementary Fig. 1a). In addition, phenotypic analyses revealed that compared with the *S. epidermidis* NC isolate, the stable SCV isolate demonstrated significantly higher biofilm formation (Fig. 2a). Moreover, proficiencies in invasion into human A549 cells and primary attachment to abiotic polystyrene surfaces were also observed in the stable SCV isolate (Fig. 2b, c), while the *S. epidermidis* Rev isolate demonstrated the same phenotypes as those of the NC (Fig. 2a–c). Since bacteria attachment is closely correlated with its physicochemical properties, we determined zeta potential and found that the stable SCV was more positive than NC, with a potential of −5.83 and −15.9 mV, respectively (Supplementary Fig. 1b, c). Thus, increased attachment of stable SCV may result

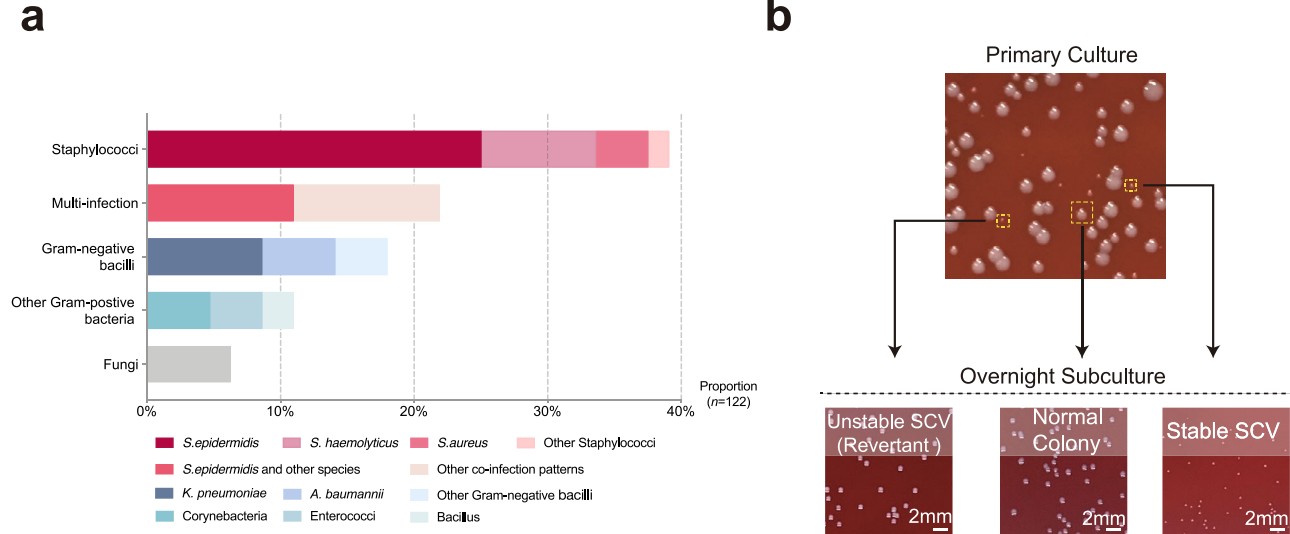

**Fig. 1 Recovery of SCV from clinical catheter infections. a** Overview of pathogens responsible for a total of 128 cases of catheter infections. Multinfection: no less than two species were isolated from the same catheter. **b** Photographs of initial culture from a representative *S. epidermidis*-infected catheter specimen after overnight incubation on sheep blood agar and subcultures of three characteristic colonies after 24-h incubation. The bars equal 2 mm.

**Table 1 Baseline features of patients with staphylococcal catheter infections.**

|  | Cases with stable SCV | Cases with unstable SCV | Cases without SCV | *P* value |
|---|---|---|---|---|
| Number of cases (%) | 8 (16%) | 33 (66%) | 9 (18%) | / |
| Female (no., %) | 2 (12.5%) | 12 (36.4%) | 4 (44.4%) | 0.7045[a] |
| Age (mean ± SD, years) | 37.2 ± 30.9 | 43.1 ± 28.1 | 54.2 ± 29.0 | 0.5860[b] |
| Infection-related symptoms (No, %)[c] | 8 (100%) | 28 (84.8%) | 7 (77.8%) | 0.3977[a] |
| Catheter-dwelling duration (mean ± SD, days) | 12.4 ± 4.5 | 11.6 ± 5.6 | 11.5 ± 8.2 | 0.9431[b] |

[a]Chi-square test.
[b]One-way ANOVA.
[c]Patients with infection-related symptoms were designated as having any abnormalities of the following parameters during the course of intubation: white blood cell count, C-reactive protein, hypersensitive C-reactive protein, serum procalcitonin, and temperature ≥ 38 °C.

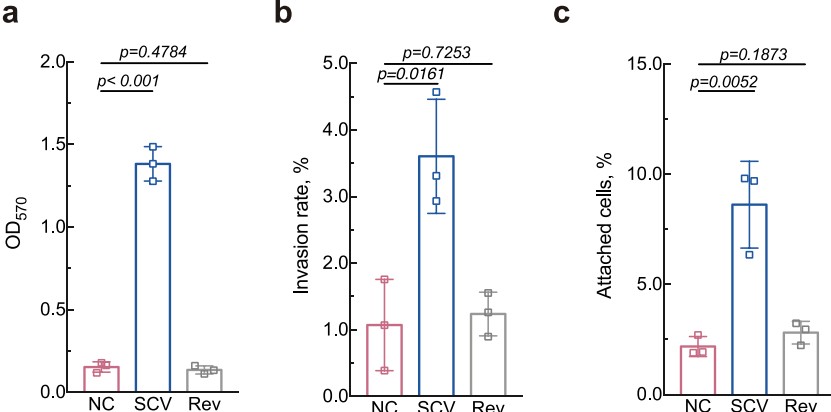

**Fig. 2 Phenotypic alterations of the *S. epidermidis* stable SCV isolate recovered from a case of clinical intravenous catheter infection. a** Biofilm elaboration measured using semiquantitative crystal violet staining. **b** Bacterial invasion assay with A549 cells. **c** Primary attachment to polystyrene surface. Statistical analyses were performed using Student's *t* test after Shapiro–Wilk normality test. The data shown are mean ± SD of three biological replicates.

from the reduction in repulsive force between stable SCV and negatively charged substratum[42].

**Genetic mutations and related metabolic disturbances of *S. epidermidis* stable SCV are involved in a wide array of cellular metabolic processes.** In view of the unusual auxotrophy exhibited by the *S. epidermidis* stable SCV isolate, whole-genome

sequencing, and untargeted comparative metabolomics analyses were then conducted to elucidate its underlying molecular events, as irreversible genetic variations are reportedly responsible for occurrence of stable SCVs[3,43,44].

Using the well-annotated *S. epidermidis* RP62A as the reference genome, the entire set of identified nonsynonymous chromosomal single-nucleotide variations based on whole-genome

sequences covered 126 genes, with 61 genes exclusive to *S. epidermidis* stable SCV, whilst there was only a subtle variance between NC and Rev in nonsynonymous single-nucleotide variation (Fig. 3a). In addition, frameshift mutations caused by stable SCV-specific indel were carried by 7 genes (Supplementary Data 1). With the purpose of systematically investigating the potential impacts of these mutations on stable SCV phenotypes, KEGG pathway analyses were then performed on the aforementioned SCV-specific mutant genes, revealing a wide range of key metabolic processes, ranging from pyruvate metabolism, citrate cycle to pyrimidine metabolism and glycine, serine, and threonine metabolism (Fig. 3b). Furthermore, there were multiple genes related to biotin synthesis, phenylalanine, tyrosine and tryptophan biosynthesis, and glycine, serine, and threonine metabolism, including 4, 2, and 2 genes, respectively (Supplementary Fig. 2a), which further implied that these metabolic pathways in *S. epidermidis* stable SCV may be disturbed by these mutations.

To ascertain whether the aforementioned genetic mutations reprogrammed the metabolism of *S. epidermidis* stable SCV, metabolic analyses were further performed on the *S. epidermidis* stable SCV and NC. The unsupervised principal component analysis of the whole collection of measured analytes indicated that *S. epidermidis* NC and stable SCV were closely clustered in biological replicates but distinctly separated from each other on the scores plot of principal component analysis (Fig. 3c), suggesting a remarkable change in stable SCV metabolism. Thirty-nine differentially expressed metabolites (DEMs), among which 28 were upregulated and 11 were downregulated in *S. epidermidis* stable SCV, were selected from the secondary metabolome set (Fig. 3d). The assignment of superclass to DEMs demonstrated that amino acids and their derivatives accounted for the majority of DEMs (16/39, 41%), followed by nucleotides and their analogs (9/39, 23%) (Fig. 3d). Moreover, contents of most detected amino acids increased significantly in *S. epidermidis* stable SCV, whereas glycine, which represents the principal ingredient of bridges linking staphylococcal peptidoglycan units, demonstrated a significantly lower level (Fig. 3d). Of note, the content of pyruvic acid, the end product of glycolysis, was significantly upregulated in *S. epidermidis* stable SCV. As well, concentration of acetyl-CoA, the central metabolite linking carbon metabolism with metabolism of fatty acids and many amino acids, was strikingly increased by the highest fold change (125-fold) (Fig. 3d).

Using the STITCH-based association network, we were able to systematically interlink mutational profile with DEMs (Fig. 3e), demonstrating that point mutations, for the most part, had limited impact on cellular metabolism. Nevertheless, frameshift mutations were highly likely to cause dramatic influences on related metabolites. In this network, glycine, whose concentration was exceptionally reduced while concentration of most DEMs were increased, was interlinked with the frameshift *SERP1287* gene (Fig. 3e), whose product is the alanine-glyoxylate aminotransferase (AGXT1) that catalyzes synthesis of glycine from alanine or glyoxylate. In addition, no mutation was found in genes involved in glycine transport, indicating that the frameshift mutation carried by AGXT1 potently accounts for the low level of glycine in *S. epidermidis* stable SCV. Further, it was also implied by the network that the dramatically increased level of acetyl-CoA was possibly attributable to mutations of *odhA* and *accA* (Fig. 3e), both of which catalyze the downstream decomposition reactions of acetyl-CoA. Pyruvate may be thus accumulated due to the inhibitory effect of acetyl-coA on the pyruvate dehydrogenase complex[45]. Despite the remarkable dysfunction of metabolome, production of endogenous reactive oxygen species (ROS), the important metabolic byproducts, was also higher in SCV as compared with NC (Supplementary Fig. 2b).

**Dysfunctional glycine biosynthesis accounts for the slow growth of *S. epidermidis* stable SCV.** Genomic and metabolomics analyses provided the evidence that slow growth of the *S. epidermidis* stable SCV isolate might result from glycine shortage, as glycine is so critical for bacterial growth that the perturbation in glycine biosynthesis would inhibit a wide variety of biosynthetic processes. In gram-positive staphylococci, glycine also serves as an integral building block of peptidoglycan cross-linking bridges whose lack can prevent peptidoglycan from being crosslinked[46], thus decelerating the cell-wall building rate. In order to verify whether the remarkable reduction in glycine level impairs growth of *S. epidermidis* stable SCV, the effect of glycine supplementation on growth of *S. epidermidis* stable SCV was next assessed.

Addition of exogenous glycine into the culture media resulted in an obvious improvement of the growth of *S. epidermidis* stable SCV at the onset of exponential phase (after 4–5 h growth), rendering $OD_{600}$ at the time of entry into stationary phase (after 12 h growth) equal to that of NC, whereas minimal effect on the growth of NC was observed (Fig. 4a). *S. epidermidis* stable SCV colonies formed on agar plates could also be restored to sizes similar to that of NC in the presence of glycine (Supplementary Fig. 2c), indicating that the *S. epidermidis* stable SCV isolate was glycine-auxotrophic SCV, an atypical kind that has not been documented so far in *S. epidermidis*.

Given that glycine serves as an ingredient of pentaglycine bridges linking staphylococci peptidoglycan units, one potential mechanism underlying the glycine-auxotrophic phenotype is deceleration of cell-wall building. The peptidoglycan-linking rates of *S. epidermidis* NC and stable SCV were thus determined by exploiting a labeling strategy based on TAMRA-bearing fluorescent D-amino acids (FDAAs) which can specifically incorporate into peptidoglycan chains. The observed remarkable reduction in TAMRA signals generated by *S. epidermidis* stable SCV over the same period of labeling time (30 min) indicated that *S. epidermidis* stable SCV has a much slower peptidoglycan-linking rate compared to NC (Fig. 4b). Nevertheless, supplementation of glycine could restore TAMRA signals from *S. epidermidis* stable SCV to the same levels as that from NC, yet there was no effect on TAMRA signals generated by NC (Fig. 4b), suggesting that shortage in glycine indeed has a negative influence on the growth of *S. epidermidis* stable SCV by decreasing the peptidoglycan-linking rate.

Taken together, these results further confirmed that the dysfunctional glycine biosynthesis pathway of *S. epidermidis* stable SCV, which contributed to a slower peptidoglycan-linking rate by minimizing ingredients integral to peptidoglycan bridges synthesis, was a decisive factor in the growth retardation of *S. epidermidis* stable SCV.

**Extracellular DNA (eDNA) released by *S. epidermidis* stable SCV promotes biofilm formation.** The observed growth defect of the *S. epidermidis* stable SCV isolate led us to propose that its enhanced biofilm formation (Fig. 2a) resulted from increased release of extracellular matrix instead of an increase in the number of cells. Thus, to illustrate the mechanism underlying the augmented biofilm formation of the *S. epidermidis* stable SCV isolate, biofilm contents were visualized using confocal laser scanning microscopy imaging (Fig. 5a). As expected, SCV gave significantly higher biofilm biomass and average biofilm thickness (Fig. 5b). Moreover, it was noticed that in contrast to NC-forming biofilm, far more propidium iodide-stained dead cells were observed in stable SCV-forming biofilm, with much fewer SYTO9[TM]-stained live cells remaining (Fig. 5b). These results suggested that the remarkable decline in cell viability of *S.*

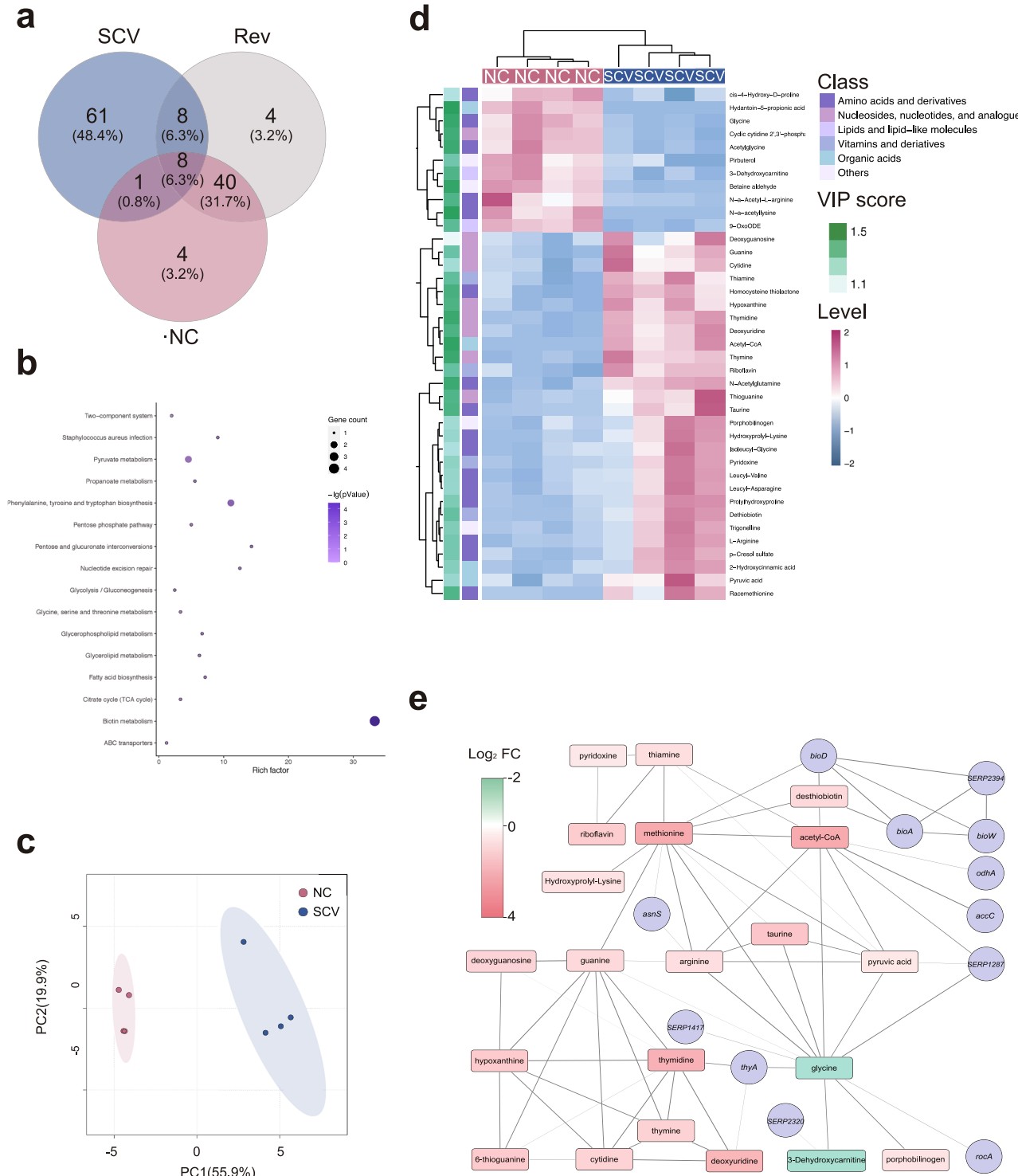

**Fig. 3 Genomic mutation-based analyses and metabolomics analyses of *S. epidermidis* stable SCV. a** Venn diagram showing numbers of SNV-containing genes overlapping between and shared by indicated isolates. **b** The Kyoto Encyclopedia of Genes and Genomes (KEGG) pathways enrichment for the mutation-related gene set. **c** Principal component analyses (PCA) based on the entire primary metabolome set. Each biological replicate is colored by group and displayed as dot. Contribution of each principal component to the total variance is given in the parenthesis as percentage. **d** A hierarchical clustering heat map illustrating normalized abundance (green lower, violet red higher), superclass, and VIP score of differentially expressed metabolites (DEMs). Each column denotes one of the biological replicates of indicated group. **e** The association network linking DEMs and mutant genes. Genes and metabolites are, respectively, shown as circles and rounded rectangles colored according to Log₂ fold change (red higher, green lower). Line thickness denotes the confidence score for the link between indicated nodes (the thicker, the higher) and only high-scoring links are shown (cutoff = 0.7).

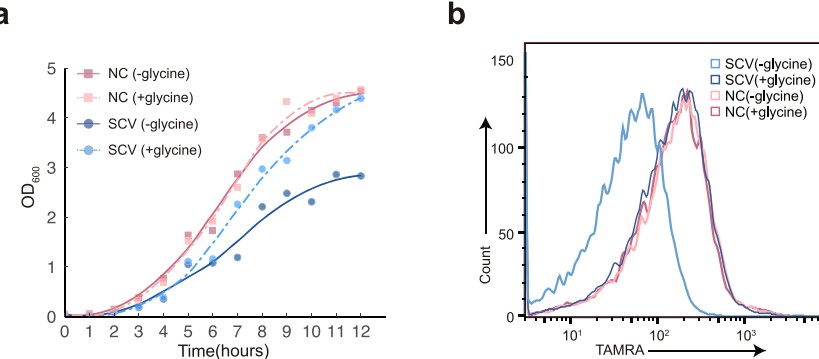

**Fig. 4 Effects of exogenous glycine on growth and peptidoglycan-linking rate of *S. epidermidis* stable SCV. a** Growth curves of *S. epidermidis* stable SCV and its normal counterpart (NC) at 37 °C in either the presence or the absence of 0.2% glycine. LOESS curve fitting was performed for all data points. Data shown are mean of two biological replicates. **b** Effects of glycine on peptidoglycan-linking rates of *S. epidermidis* stable SCV and its NC. Bacterial cells grown in the presence or absence of 0.2% glycine were labeled by TAMRA-bearing FDAA probes over half an hour period and then subjected to flow cytometry analyses.

*epidermidis* stable SCV may influence biofilm matrix composition by facilitating intracellular DNA release. Semiquantitative analysis of eDNA by extracting eDNA from stable SCV-forming biofilm further verified that eDNA, particularly high-molecular weight eDNA, was abundantly released by stable SCV (Fig. 5c). However, there was no significant change in the production of PIA, the widely recognized predominant biofilm matrix of *S. epidermidis*[47], as reflected by both fluorescent signals of Alexa Fluor 350-conjugated wheat-germ agglutinin (WGA) and semi-quantitation (Fig. 5d). Moreover, *icaA*, whose product synthesizes PIA oligomers, did not display a significantly higher transcriptional level (Supplementary Fig. 3a). Also, the *S. epidermidis* stable SCV isolate shared a similar decline in biofilm biomass with NC after protease K treatment, whereas after being treated with DNase I, the biofilm biomass of *S. epidermidis* stable SCV decreased by a significantly higher proportion than that of NC (Fig. 5e), further illustrating that it is the increased release of eDNA that directly potentiates biofilm elaboration of *S. epidermidis* stable SCV.

Considering that cell death of *S. epidermidis* stable SCV spontaneously occurred under biofilm condition, we posited increased cell death of SCV is due to upregulated cell lysis. To verify whether the dramatic decline in cell viability is due to upregulation of autolysis activity, Triton-X100-induced autolysis rates of *S. epidermidis* stable SCV and NC cells grown in TSB supplemented with 0.5% glucose (TSBg) medium were determined over a period of 5 h. As expected, *S. epidermidis* stable SCV demonstrated a distinguishably higher Triton-X100-induced autolysis rate (Fig. 6a). Moreover, determination of eDNA concentration at different time points revealed that *S. epidermidis* stable SCV starts to accumulate much eDNA than NC at the initial stage of exponential growth stage, and concentration of eDNA is constantly growing during all later growth stages, while production of eDNA in NC remains much lower during the entire course of growth (Fig. 6b). Analysis of cell viability by flow cytometry further verified that *S. epidermidis* stable SCV continuously undergoes autolysis from exponential growth phase to stationary growth phase, though autolysis of exponentially growing cultures is more faster than that of cultures at stationary stage, yet still at a much quicker speed than NC (Supplementary Fig. 3b). These results collectively validated that it is the rapid autolysis that contributes to the observed significant decline in cell viability and subsequent DNA release.

However, the underlying event responsible for the rapid autolysis rate was not due to upregulation of autolysin, as the transcriptional level of the autolysin-encoding gene *atlE* was not significantly upregulated (Supplementary Fig. 3a), whereas

effectors of another autolysis-related system, namely *cidA* and *lrgA*, were upregulated and downregulated 2.6-fold and 4.7-fold, respectively (Fig. 6c). *lrgA* and *cidA* are functionally antithetical effectors wherein *cidA*-encoded CidA induces cell lysis, while *lrgA*-encoded LrgA prevents this process[48], indicating that the rapid autolysis of *S. epidermidis* stable SCV was triggered probably as a result of the imbalance between *cidA* and *lrgA* expression.

Notably, the contrasts of both autolysis rate and biofilm-forming ability between *S. epidermidis* NC and stable SCV were attenuated when additional glucose was eliminated from TSBg medium (Fig. 6a, d), leading us to reason that excessive glucose is a key factor mediating the stronger autolysis rate of *S. epidermidis* stable SCV. Overflowed glucose is reportedly able to increase build-ups of acetic acid[49], thus concentrations of acetate released into supernatant in either the presence or the absence of excessive glucose were assayed. Consistently, an increase in acetate concentration under the condition of overflowed glucose was observed (Fig. 6e). Also of note was the small but significantly higher build-up of acetic acid in *S. epidermidis* stable SCV (Fig. 6e), which is probably a consequence of its relatively higher levels of pyruvate and acetyl-CoA, as revealed by LC/MS–MS. However, when evaluating the roles of acetic acid in autolysis activity, antithetical effects toward *S. epidermidis* NC and stable SCV were observed. It was found that acetic acid potentiated autolysis of *S. epidermidis* stable SCV, but on the contrary, it inhibited autolysis activity of NC (Fig. 6f). Their distinct responses to acetic acid were then found to be a potential result of different responsiveness of *lrgA* and *cidA*. Acetic acid could remarkably activate *lrgA* expression of NC, but hardly activated *lrgA* of *S. epidermidis* stable SCV (Fig. 6g). In contrast, no significant effect on *cidA* was observed for NC, whereas *cidA* expression of *S. epidermidis* stable SCV was upregulated in the presence of acetic acid (Fig. 6g). Consistent with these results was another finding demonstrating that the autolysis activity under the condition of excess glucose could be reversed by the chemical pH-buffering agent HEPES, which buffered culture medium to a pH of 7.0 (Fig. 6h). Since acetic acid is a typical weak acid with a pKa value of 4.7 and can permeate into cells freely only if in its deionized form, its deionization and free access to the interiors of cells would be limited if culture medium was buffered by pH-buffering agents such as HEPES to neutral pH. Accordingly, the autolysis-driven biofilm formation of *S. epidermidis* stable SCV was measurably attenuated by exogenous addition of HEPES, but there was no inhibitory effect on that of NC (Fig. 6i), further indicating that the distinctive autolysis-driven biofilm-forming

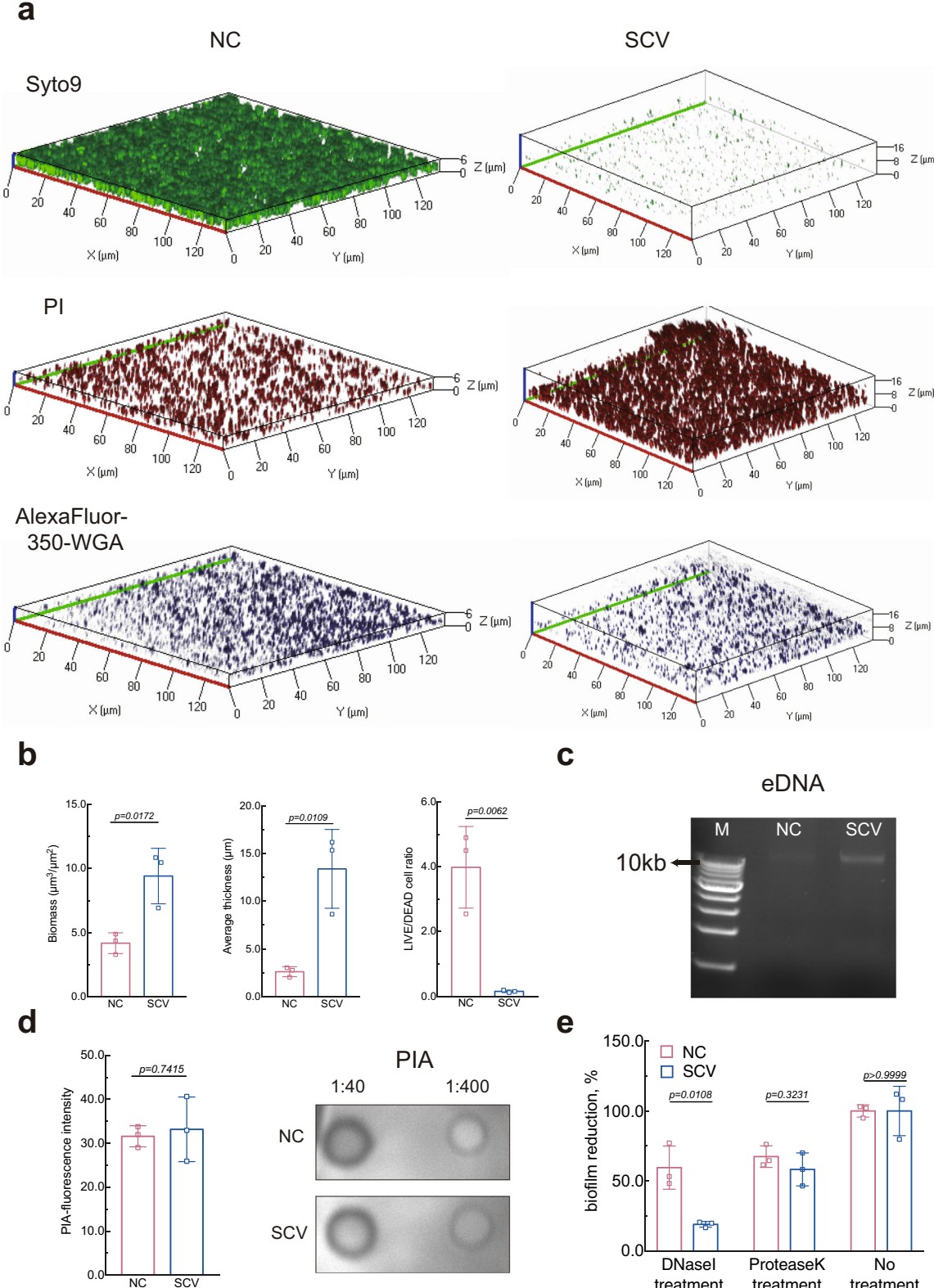

ability of *S. epidermidis* stable SCV was, at least in part, attributable to acetic acid, which is able to potentiate cell death and lysis of stable SCV but inhibits autolysis activity of NC.

**S. epidermidis stable SCV facilitates communal survival in catheter-related infection.** Despite the observed decline in cell

viability, *S. epidermidis* stable SCV was able to offer survival advantages to benefit the entire population. The effects of *S. epidermidis* stable SCV at a range of initial inoculum doses on in vitro NC biofilm formation were considerable, since the addition of stable SCV could significantly increase NC biofilm biomass even at a negligible slightly small inoculum (Fig. 7a). More importantly, when *S. epidermidis* stable SCV and NC were

**Fig. 5 Increased release of extracellular DNA (eDNA) facilitates biofilm formation of *S. epidermidis* stable SCV. a** Representative confocal laser scanning microscopy images of biofilms formed by *S. epidermidis* stable SCV and normal counterpart (NC). Biofilm were grown in TSBg for 24 h and then stained with propidium iodide (PI, for dead cells), SYTO9 (for live cells), and WGA-Alexa Fluor^TM 350 (for PIA). **b** Analysis of the confocal laser scanning microscopy z-stack images using COMSTAT2 (http://comstat.dk) and ImageJ software (https://imagej.nih.gov/ij). Data shown are mean ± SD from three technical replicates. **c** Semiquantitative measurement of eDNA released by *S. epidermidis* stable SCV and NC into biofilm. M: 1 kb marker. **d** Mean fluorescence intensity of PIA within biofilm and semiquantification of PIA retrieved from biofilm formed by *S. epidermidis* stable SCV and NC. The numbers shown are dilution factors of PIA extracts. Data shown are mean ± SD from three technical replicates. **e** Effects of DNase I and protease K on biofilm biomass. Data shown are relative optical density at 570 nm of biofilm biomass after treatment as indicated with DNase I or protease K or none of both. Data shown are mean ± SD from three biological replicates. Unpaired Student's *t* test was used for statistical analyses after Shapiro–Wilk normality test.

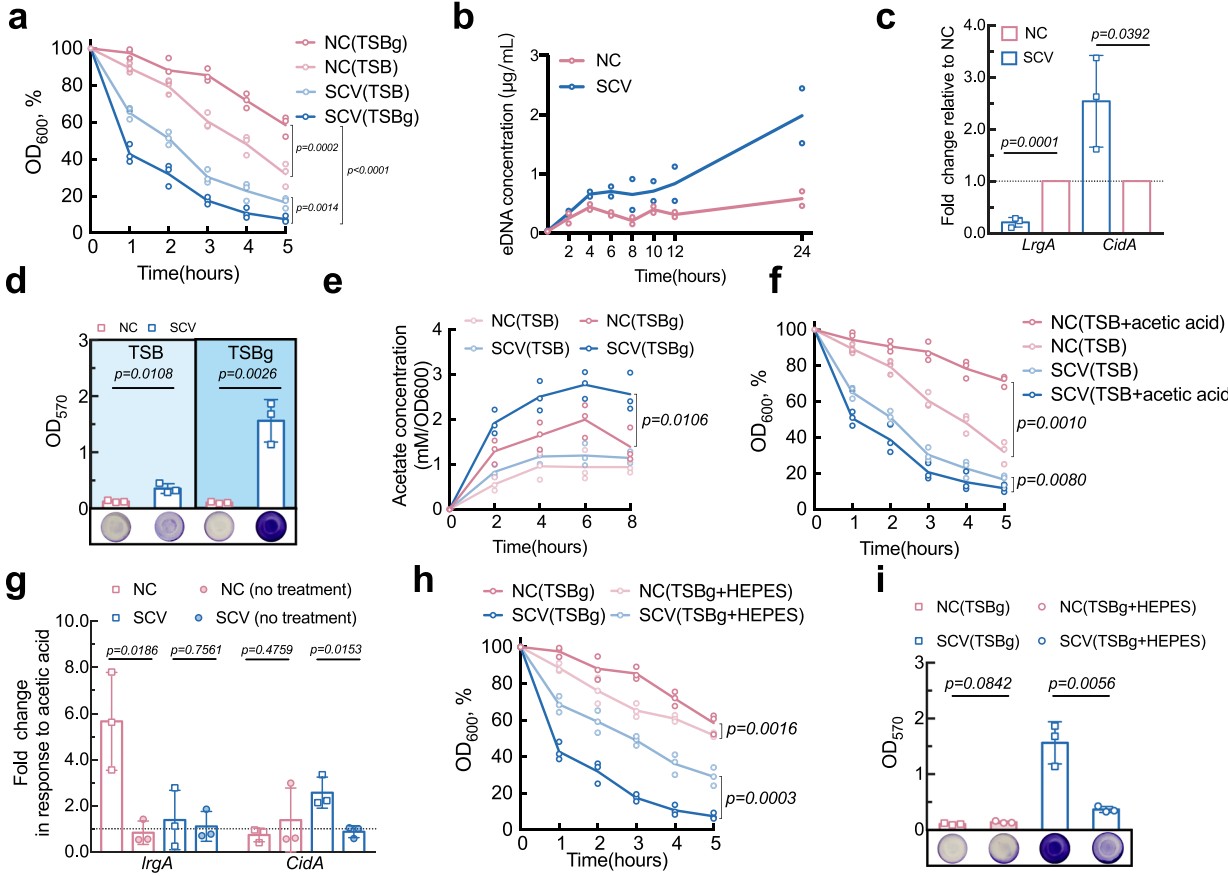

**Fig. 6 *S. epidermidis* stable SCV generates much more acetic acid that triggers autolysis of SCV cells but inhibits autolysis of NC cells. a** Triton-X-100-induced autolysis activity. Bacterial cells were grown in TSB either without or with 0.5% glucose (TSBg). Statistical analysis indicated in the graph was performed by two-way ANOVA with Bonferroni's multiple comparison post-test. **b** Concentration of cell-free eDNA from batch cultures at indicated time points. Data shown are two biological replicates. **c** Transcriptional level of *lrgA* and *cidA* determined by quantitative real-time RT-PCR using primers listed in Supplementary Table 2. The expression of *gyrB* gene was used for normalization. Unpaired Student's *t* test was used for statistical analyses after Shapiro–Wilk normality test. **d** Semiquantification of biofilm formed by NC and stable SCV grown in TSB either without or with 0.5% glucose. Unpaired Student's *t* test was used for statistical analyses after Shapiro–Wilk normality test. **e** Acetate concentrations of cultures grown in TSB supplemented with or without 0.5% glucose. Concentrations of supernatant acetate were determined according to instructions of the reagent kit and normalized according to OD_{600}. Statistical analysis indicated in the graph was performed by two-way ANOVA with Bonferroni's multiple comparison post-test. **f** Triton-X-100-induced autolysis activity of NC and stable SCV cells grown in TSB without or with acetic acid (30 mM, pH 4.8). Statistical analysis indicated in the graph was performed by two-way ANOVA with Bonferroni's multiple comparison post-test. **g** Relative fold changes of *lrgA* and *cidA* in response to 30 mM acetic acid. The expression of *gyrB* gene was used for normalization. Unpaired Student's *t* test was used for statistical analyses after Shapiro–Wilk normality test. **h** Effects of the buffering agent HEPES on Triton-X-100-induced autolysis activity. Statistical analysis indicated in the graph was performed by two-way ANOVA with Bonferroni's multiple comparison post-test. **i** Effects of the buffering agent HEPES on biofilm formation. Unpaired Student's *t* test was used for statistical analyses after Shapiro–Wilk normality test. Data shown are three biological replicates unless otherwise specified. Mean values of each group at indicated time points are connected by solid line.

coinfected in a mouse catheter-infection model, the CFU of bacteria recovered from implanted catheters and surrounding tissues was significantly higher, compared to that from independent infection of either *S. epidermidis* stable SCV or NC (Fig. 7b).

Paradoxically, despite the fact that *S. epidermidis* stable SCV demonstrated higher autolysis rate in vitro, it did not generate lower CFU than NC during biofilm infection in vivo (Fig. 7b), supposedly because its augmented biofilm protects it from immune killing more potently, while NC was more amenable to

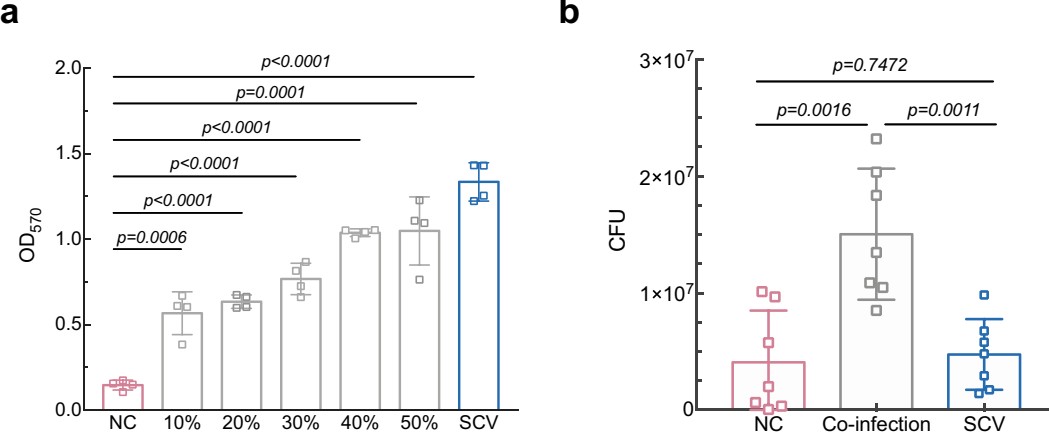

**Fig. 7 Beneficial effect of *S. epidermidis* stable SCV subpopulation. a** Effect of *S. epidermidis* stable SCV subpopulation on in vitro biofilm formation of NC. Culture medium was inoculated with a mixture of NC and SCV to a final $OD_{600}$ of 0.1 and the percentage of SCV in the inocula is indicated. Data shown are mean ± SD from four biological replicates. Unpaired Student's *t* test was used for statistical analyses after Shapiro–Wilk normality test. **b** Mutual benefits of stable SCV in mice subcutaneous catheter infections. The mouse model was performed with $n = 7$ per infection pattern. Challenging catheters for coinfection was prepared by incubation in mixed inocula of NC and SCV at a ratio of 3:2. CFU recovered from the catheters and adjacent skin tissues were counted on day 5 by plating onto sheep blood agar plates after sonication and vortex of catheters and homogenization of skin tissue samples. Error bars show mean ± SD. Unpaired Student's *t* test was used for statistical analyses after Shapiro–Wilk normality test.

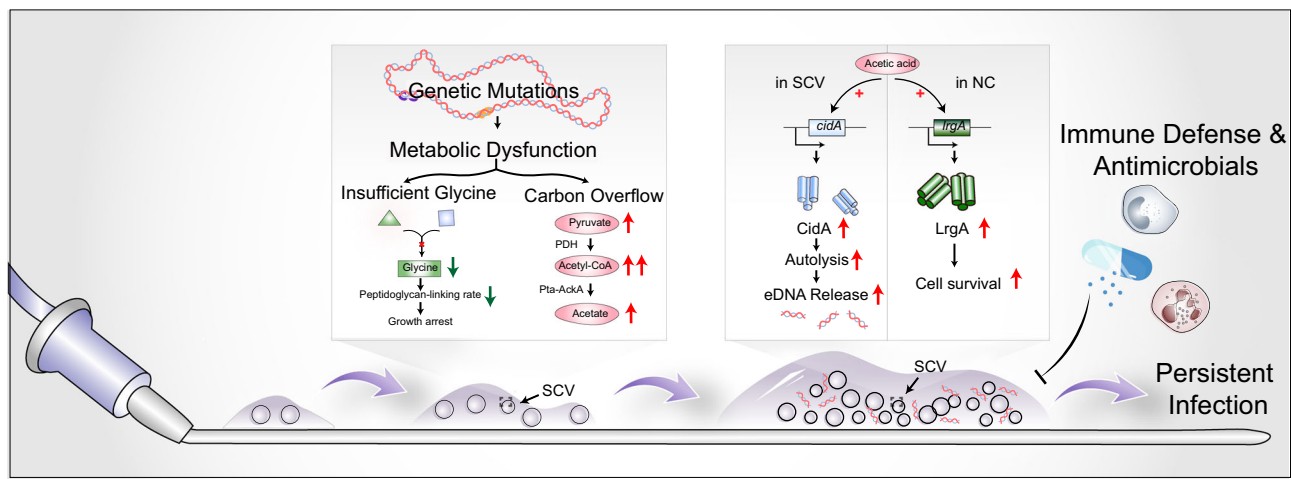

**Fig. 8 Working model of the glycine-auxotrophic clinical *S. epidermidis* SCV isolate in the persistence of biofilm-related catheter infections.** Under the exposure of antibiotics, SCV subpopulations emerge rapidly, accompanied by genomic variation and consequent metabolic dysfunction. Frameshift of AGXT1 in SCV potentially results in a defective glycine biosynthetic pathway, which leads to slow cell-wall building rate and consequent growth arrest. Overflowed pyruvic acid and acetyl-CoA lead SCV to generate and release a higher level of acetic acid, which triggers cell autolysis of SCV cells but protects NC cells against lysis by upregulating *cidA* expression of SCV and *lrgA* expression of NC, thereby allowing SCV to far more quickly autolyze. Rapid autolysis of SCV leads to the leakage of intracellular contents such as DNA and enhances biofilm formation, which renders the whole population recalcitrant toward both host immune system and antimicrobial agents and more importantly, facilitates persistent infection.

host defense. Accordingly, *S. epidermidis* stable SCV and its NCs compensated for each other during in vivo infection, wherein SCV enhanced biofilm formation to keep NC from both immune and antibiotic killing, thus viable NC cells would cause infection relapse or persistence once antibiotic therapy was halted or the immune system was suppressed. Therefore, these data implicated that SCV plays an altruistic role in promoting the occurrence and persistence of biofilm-related catheter infections.

## Discussion

Collectively, the present work deciphers a unique and intriguing strategy exploited by a glycine-auxotrophic clinical *S. epidermidis* stable SCV isolate in the persistence of biofilm-related infections (Fig. 8). SCV subpopulations accompanied by genomic variation and consequent metabolic dysfunction emerge during the course

of infection. Frameshift AGXT1 in SCV potentially results in a defective glycine biosynthetic pathway, which leads to a slow cell-wall building rate and consequent growth arrest. Due to carbon overflow, SCV generates and releases a higher level of acetic acid, which triggers cell autolysis of SCV cells but protects NC cells against lysis by upregulating *cidA* expression of SCV and *lrgA* expression of NC, thereby allowing SCV to far more quickly autolyze, release large quantities of eDNA, and more importantly, facilitate in-host communal survival.

Biofilm formation is a key process in the occurrence and development of catheter infections, as well as many other types of foreign body infections[50]. *S. epidermidis* is hardwired to elaborate biofilm and the emergence of a SCVs population could, as revealed here, dramatically improve biofilm formation in an altruistic manner. Obviously, the biofilm-forming capability of

the *S. epidermidis* stable SCV isolate was even higher in the presence of excess glucose, which is of great clinical importance in the context of intravascular catheter infections, especially for hyperglycemic patients[51]. During the course of biofilm formation, *S. epidermidis* stable SCV rapidly autolyzes in the interests of the entire population and generates beneficial products, namely eDNA, which provides not only adhesive forces for adherence to catheter surfaces but also strong cohesiveness for aggregation into multicellular structures[52]. Nevertheless, for many years, there has been some doubt about the role of eDNA in *S. epidermidis* biofilm development, due to the notion that it is PIA rather than eDNA that has long been recognized as the predominant matrix molecule for biofilm formation of *S. epidermidis*[53]. However, recognition of the important role of eDNA in biofilm elaboration is continuing to gain ground and a growing body of evidence has revealed the significance of eDNA in *S. epidermidis* biofilm[54,55].

Release of eDNA into biofilm is largely dependent of cell death, which is either programmed (i.e., programmed cell death) or triggered by external stimuli[56]. Programmed cell death is generally considered to be integral for multicellular organisms but undesirable for unicellular microorganisms, as autolysis delivers a fatal blow to single-celled bacteria, let alone confers any evolutionary benefits. Yet the structurally elaborated biofilm, which embeds not only highly differentiated bacteria but also a series of functionally distinguished biomolecules, is a typical of multicellular community. Further, there is also increasing evidence indicating that programmed cell death is a critical part of bacterial biofilm development. However, programmed cell death is mediated by a battery of coordinated effectors among which CidA and LrgA are tightly under the control of metabolic signals, such as carbon overflow[57]. Though originally identified in *S. aureus*, CidA and LrgA proteins are found to be conserved in several other species including *S. epidermidis*[54,58,59]. As CidA shares higher structural similarity with the bacteriophage-encoded holin protein, it was proposed to induce cell lysis and death in a manner similar with holin[48,60]. In contrast to CidA which enhances murein hydrolase activity, however, another protein functionally antithetical to CidA, namely LrgA, is proposedly able to prevent the access of murein hydrolases to the peptidoglycan, thus inhibiting cell lysis[48]. Yet, it was earlier clarified that CidA may not function as a prolytic protein as it was originally reported[61]. Nevertheless, a recent investigation indicated that CidA is a crucial effector mediating mupirocin-induced cell lysis and death in the USA300 *S. aureus* strain[56], suggesting that the functional significance of CidA in determining cell lysis and death may vary from strains to strains. Furthermore, inconsistent with previous findings which suggested that acetic acid generated under glucose overflow conditions is able to induce expression of both *cidA* and *lrgA* of *S. aureus*[49], no significant effect of acetic acid on *cidA* transcriptional level of the *S. epidermidis* NC strain was observed in this work. Considering that effectiveness of a weak acid is dependent on both successful diffusion across the cell membrane and intracellular dissociation, an alternative explanation for this inconsistency is the variances between strains in membrane potential and permeability, pH gradient across the membrane, intracellular pH, etc. Thus, extracellular acetic acid at such concentration may not be able to activate intracellular regulator CidR of *S. epidermidis* NC, whereas LrgA, whose regulatory system LytRS can sensitively respond to environmental triggers[62], is upregulated in the presence of acetic acid. Also, the dysfunctional metabolic state in *S. epidermidis* stable SCV, which potentially contributes to alterations in membrane properties and intracellular pH homeostasis, may drive different outcomes for *S. epidermidis* stable SCV and NC in response to acetic acid. However, the precise functions and regulatory work of Lrg/Cid still remain unclear and controversial. Therefore, further

verification is still needed to fully understand the functional roles and regulatory networks of the Lrg/Cid system in *S. epidermidis*.

Nonetheless, the underlying causes for the rapid autolysis of *S. epidermidis* stable SCV are seemingly multifaceted. As can be seen from the reduction in autolysis speed of bacterial cultures at stationary growth phase, specific growth rate is also a potential factor determining individual cell fate. Exponentially growing bacterial cells typically have much higher growth speed, or more specifically, more active peptidoglycan synthesis, which involves pentaglycine bridges-mediated cross-linking. In such a context, *S. epidermidis* stable SCV with inefficient synthesis of pentaglycine would form aberrant cross-linking bridge and subsequently, abnormal pentaglycine interpeptide bridges are able to potentiate cell lysis. Also, metabolic dysfunction is another potential driver for the continuous autolysis during the course of growth. In addition to acetic acid, overproduction of other metabolic intermediate products, such as ROS, can impair intracellular biological activities and affect whole cell physiology. Though sometimes immune protective, ROS would be deleterious if excessively generated[63]. Thus, higher levels of intracellular ROS accumulation in *S. epidermidis* stable SCV can also promote its cell lysis and death[64].

Even though metabolic alterations always get entangled in the SCVs formation, there are limited analyses based on the metabolic profile of *S. epidermidis* SCVs so far. With the help of LC–MS/MS, a reliable and compatible technique to sensitively detect small-molecule metabolites, we were able to identify the glycine-auxotrophic *S. epidermidis* SCV, which has not been documented hitherto. This strain demonstrated no auxotrophy of either hemin or menadione, both of which were frequently reported in ETC-deficient SCVs, partly because it carried no mutations in genes involved in hemin or menadione biosynthesis. A previous report showed that the growth rate of menadione-auxotrophic *S. aureus* SCV was not affected by the addition of a series of L-amino acids[65], indicating that glycine shortage inhibited bacterial growth via another pathway distinct from menadione-related cell respiration, or rather, the ETC. Glycine contributes substantially to bacterial growth by not only serving as protein building blocks but also influencing a variety of biosynthetic processes, such as peptidoglycan linking[46]. As the major ingredient of pentaglycine bridges linking peptidoglycan units, glycine plays a functional role in cell-wall biosynthesis[66]. The characteristic pentaglycine bridge in staphylococci allows peptidoglycan to form a dense, rigid, and three-dimensional cell-wall structure necessary for cell division and growth, thus normal peptidoglycan assembly relies on glycine availability, while glycine depletion would shorten the bridges and generate more nonbridge-linked peptidoglycan units[46]. Also, it was observed that repression of *fmhB*, whose product catalyzes the first step in the synthesis of pentaglycine interpeptide, was able to cause growth arrest of *S. aureus*[67]. Moreover, elimination of glycine from culture media was found to decelerate exponential growth of *S. aureus*[46], further indicating a crucial role of glycine in staphylococcal growth. However, the effects of glycine on bacterial growth appeared to be dose dependent, as high concentrations of glycine could inhibit growth by interfering with other peptidoglycan synthetic steps[68,69], thus the effect of glycine supplementation on the growth of NC was simultaneously assessed when testing growth of stable SCV under the usual glycine concentration (200 mg/L). Our results suggested that addition of 0.2% glycine into culture medium had no obvious influence on the growth of normal-sized *S. epidermidis* but started to accelerate growth of *S. epidermidis* stable SCV after 4 h of growth, which is consistent with the fact that staphylococci begin to utilize exogenous glycine at the time of entry into the exponential phase[46]. Furthermore, FDAAs, which were initially designed to investigate

transpeptidation (cross-linking of the pentaglycine bridge to the D-Ala in position four of a neighboring stem) involved in the peptidoglycan biosynthesis[70], are a powerful indicator of cross-linking activities, thereby rendering cross-linking rates comparable. The significant reduction in cross-linking rate of *S. epidermidis* stable SCV further indicated that glycine insufficiency can impair growth by decelerating cell-wall linking.

Considering that exogenous glycine is able to restore the growth of stable SCV, glycine shortage in *S. epidermidis* stable SCV is in all likelihood due to a biosynthetic defect rather than inactivation of glycine transporter. The disturbed glycine biosynthesis of *S. epidermidis* stable SCV was most probably due to the frameshift mutation in *SERP1287*, whose product belongs to the AGXT family. Despite the fact that the function of this aminotransferase in *S. epidermidis* is not fully characterized, it has already been confirmed that AGXT can synthesize glycine from glyoxylate or alanine[71], which represents an important glycine biosynthetic route in addition to serine and choline transformation. Unfortunately, further validation of the causality between AGXT mutation and glycine shortage was strongly impeded by the considerable restriction-modification barriers that existed in this clinical isolate, which rendered successful electroporation intractable.

To the best of our knowledge, it is the first study revealing that the clinically isolated glycine-auxotrophic *S. epidermidis* SCV and its NC can act in concert to facilitate occurrence and persistence of biofilm-related catheter infections. Be that as it may, the intricacy of mutational and metabolic profile and the diversity of pathological mechanism in clinical *S. epidermidis* SCVs indicate that *S. epidermidis* SCVs remain an issue far from being solved, which requires more in-depth investigations.

## Methods

**Culture-based analyses of catheter infections**. Catheter specimens were collected from six different medical centers in Shanghai, China, for a period of 6 months. Catheters were submerged in 2 mL fresh TSB liquid medium and vortexed for 15 min after sonication for 5 min. Appropriate dilutions or concentrates from each specimen were then plated on 5% sheep blood agar and incubated at 37 °C for at least 24 h. For plates with colonies of similar colony morphology, ten colonies were randomly selected for species identification. For plates with colonies of apparently different colony morphologies, three colonies of each colony morphology were selected and subjected to species identification using MALDI-TOF-MS (Bruker Daltonics, Bremen, Germany) according to the manufacturer's instructions. However, as to plates with SCVs, which have apparently smaller colony size than but share other morphological features with its NC, at least 50 SCVs from the specimen were streaked onto another 5% sheep blood agar plate, then incubated for 24 h, and simultaneously each plate was streaked with a homologous normal colony to facilitate further stability analysis of SCVs. SCVs that had reverted to the same size as their NC were classified as unstable SCVs, while SCVs that remained at a smaller colony size than their NC were regarded as stable SCVs. All SCVs have undergone species identification and only those of the same species as their NC were included for subsequent investigations.

**Animal experiments**. A mouse model of subcutaneous catheter infection was constructed. Briefly, 1-cm catheter pieces were incubated within early log-phase bacterial culture for 2 h at 37 °C. Then, all catheters were taken out and gently rinsed with PBS three times. Dried catheters were then inserted under the dorsal skin of 4- to 6-week female BALB/c mice. CFU were harvested from catheters and homogenized surrounding skin tissue.

**Semiquantification of biofilms and observation by CLSM**. Crystal violet staining was applied to semiquantify biofilm formation of *S. epidermidis* isolates. Briefly, overnight bacterial cultures were diluted into TSBg to a final optical density of 0.1. The diluted cultures were aliquoted to 96-well flat-bottom tissue culture plates (200 μL/well) and statically incubated at 37 °C for 24 h. Wells were washed with PBS after gentle removal of culture supernatants. Bouin's fixative was added onto the bottom of the wells to treat biofilm for 1 h. The fixative was gently aspirated out and wells were washed three times with PBS, and stained with 0.4% (wt/vol) crystal violet dissolved in 1% ammonium oxalate solution containing 20% absolute ethanol. Biofilm formation was measured by the Synergy H1 plate reader (BioTeK, USA) at 570 nm after being washed by PBS to remove redundant crystal violet. To compare the relative abundance of DNA and protein in SCV-forming and NC-forming biofilm, Dnase I (Yeasen, 10608ES80) was added at the beginning of 24-h

incubation, while protease K (Yeasen, 10411ES80) was added onto established biofilm after supernatant was discarded. Biofilm formation was then quantified as described above. For biofilm observation by CLSM, biofilm grown in glass-bottom dish (NEST, 801001) was washed with PBS three times, and then sequentially stained with 1 μM of SYTO9 (Invitrogen) for 30 min, 1 μM of propidium iodide (Yeasen, Shanghai) for 15 min, and 2.5 μg/mL Wheat Germ Agglutinin-Alexa Fluor™ 350 conjugate (Invitrogen, W11263) for 20 min. The stained cells and PIA were visualized by Zeiss LSM 880 upright laser scanning confocal microscope with a 63×/1.4 oil immersion objective. Three-dimensional biofilm images were generated by ZEN lite.

**Cell invasion assay**. At least 2 h before the challenge, antibiotic containing media of tissue culture cells (TCC) A549 (from laboratory stock) were replaced with F12K/DMEM without antibiotics ($2 \times 10^5$ cells per well for the 24-well plate). Overnight grown bacterial cultures in TSB were diluted into 10 mL TSB (1:100 dilution). Diluted bacterial cultures were then incubated at 37 °C on a rotating wheel for 3 h. Bacteria cultures were washed with fresh F12K/DMEM media and $OD_{600}$ were adjusted to 0.4. Each well of TCC was washed with PBS twice, and cells were infected with 500 μL of bacterial cultures at MOI of 100 bacteria per cell for 2 h at 37 °C. At this point appropriate dilutions of the bacterial cultures were prepared to count the bacterial inoculum. After 2 h of infection, TCC were washed twice with PBS and lysed with 0.5 mL of PBS containing 0.1 % sodium decholate, then the TCC were mechanically disrupted with a micropipette tip by pipetting up and down. Appropriate dilutions were made and plated onto TSA plates to count intracellular bacteria after overnight incubation. The invasion rate was determined as the percentage of CFU counts of intracellular bacteria among CFU counts of bacteria that were initially added into cells.

**Primary attachment assay**. Overnight cultures of *S. epidermidis* strains were diluted into TSBg by a factor of $2.5 \times 10^7$. The diluted bacterial cultures were aliquoted to a 12-well polystyrene flat-bottom plate (Corning, 3513), with a final volume of 1 mL and then incubated at 37 °C for an hour. Subsequently, the spent media supernatant was aspirated gently out of each well. The well was then washed with sterile PBS three times to remove unattached cells. Prewarmed TSA (TSB supplemented with 0.8% agar) was added into each well after careful washing. The number of attached cells was counted after overnight culturing of the 12-well plate. Attachment proportion was calculated by dividing the number of attached cells by CFU of the initial $2.5 \times 10^7$-fold dilution, which was also calculated after overnight culturing on a TSA plate.

**Zeta potential measurement**. The overnight grown bacterial cultures were centrifuged at 14000 rpm for 20 min and washed five times with PBS. The OD600 of the bacterial suspensions were adjusted to 0.2. Then, 1 mL of the prepared bacterial suspension was added to 2 mL PBS. The mixture was then transferred the Malvern polystyrene U-shaped cell after being vortexed and subjected to zeta potential measurement using Zetasizer (Malvern Panalytical Ltd).

**Whole-genome sequencing**. Total genomic DNA of each strain was respectively extracted using standard phenol–chloroform method. Purity and concentration of DNA samples was monitored on 1% agarose gels. For qualified genomic DNA, sequencing libraries were constructed and quality checked. Pair-end sequencing was performed on an Illumina® HiSeq platform, with a read length of 150 bp at each end. Sequenced reads were filtered by trimming adapter and discarding paired reads that contain more than 10% uncertain nucleotides or more than 50% low-quality nucleotides (base quality < 5) of either read. Filtered clean reads were then mapped to the *S. epidermidis* RP62A genome by BWA software (http://bio-bwa.sourceforge.net/bwa.shtml). SAMtools (http://samtools.sourceforge.net/) and Picard (http://picard.sourceforge.net/) were, respectively, used to sort alignments and mark duplicate reads. SNP detection and annotation were performed using GATK and SnpEff, respectively, with the default parameter. Indels were detected using the same method.

**Metabolic profiling**. Untargeted metabolomics analyses of *S. epidermidis* were carried out by LC–MS/MS analyses on an UHPLC system (Vanquish, Thermo Fisher Scientific) with a UPLC BEH Amide column (2.1 mm × 100 mm, 1.7 μm) coupled to Q Exactive HFX mass spectrometer (Orbitrap MS, Thermo Fisher Scientific). In brief, metabolites of each strain were extracted by harvesting exponential cultures grown in TSBg, repeatedly homogenizing and sonicating with the addition of extract solution (acetonitrile:methanol = 1:1, containing isotopically labeled internal standard mixture). LC–MS/MS analysis was then conducted for metabolites extracts. An in-house MS2 database was applied in metabolite annotation after peaks were detected, extracted, aligned, and integrated. Data generated from both positive ion mode and negative ion mode were integrated for downstream analyses after filtration, transformation, and normalization. The final dataset containing the information of peak number, sample name, and normalized peak area was imported to the SIMCA15.0.2 software package (Sartorius Stedim Data Analytics AB, Umea, Sweden) for multivariate analysis. Principle component analysis, an unsupervised analysis that reduces the dimension of the data, was carried out to visualize the distribution and the grouping of the samples using the

Web-based platform MetaboAnalyst (http://www.metaboanalyst.ca). Furthermore, the value of variable importance in the projection (VIP) of the first principal component in OPLS-DA analysis was obtained. The metabolites with VIP > 1, $p$ < 0.05 (Student's $t$ test), and fold change > 2 or <0.5 were considered as significantly changed metabolites. The acetate assay kit (Abcam, 204719) was used for determination of acetate concentration according to manufacturer's instructions, and final results were normalized by the $OD_{600}$.

**Determination of cellular ROS**. The cell permeant reagent 2′,7′-dichlorofluorescin diacetate included in the Cellular ROS Assay Kit/Reactive Oxygen Species Assay Kit (Abcam, 113851) was used to quantitatively assess ROS in bacterial samples. In brief, single colony from sheep blood agar plate was inoculated into TSB and incubated at 37 °C, 200 rpm for 12 h. Then, 20 μM 2′,7′-dichlorofluorescin diacetate was mixed with bacterial cells in the buffer supplied by the manufacturer. After incubating for 1 h at 37 °C, bacteria were washed by PBS and then subjected to flow cytometry using the BD FACSverse system (BD Biosciences, San Jose, CA, USA). FlowJo software (Version 10.4) was used for data visualization.

**In vitro labeling with FDAA probes**. Single colonies grown on sheep blood agar plates were streaked into LB broth, which was supplemented with 0.2% glycine if indicated. After overnight incubation, cultures were diluted by 100-fold into fresh LB broth supplemented with or without 0.2% glycine and then incubated at 37 °C under shaking conditions. After 2 h of incubation, FDAA probes (Chinese Peptide Company, 979938) were added to the cultures to a final concentration of 0.3 mM. The bacteria were then washed twice with PBS after being labeled for half an hour at 37 °C, 200 rpm. Cytometry analyses were performed on BD FACSuite software and BD FACSverse system (BD Biosciences, San Jose, CA, USA). For each sample, 10,000 events were collected. FlowJo software (Version 10.4) was used for data visualization.

**Quantification of high-molecular weight eDNA in biofilm**. Biofilm formation and quantification of high-molecular weight eDNA in biofilm was performed[72]. Briefly, culture supernatant was then discarded and 1 mL PBS was added to the well. Contents were transferred to another tube using a cell scraper (NEST, 710001). After centrifugation, supernatant was removed and the cell pellet was thoroughly suspended in PBS and subjected to drastic vortex. Five microliters of the cell pellet were assessed for the presence of eDNA using 1% agarose gel electrophoresis.

**Quantification of PIA in biofilms**. A dot blot assay with a WGA-horseradish peroxidase (HRP) conjugate (Sigma-Aldrich, L3892) was performed for PIA quantitative measurement. In brief, biofilm was firstly grown as mentioned above and subsequently scraped off the surface by a cell scraper (NEST, 710001). Then, the sediments were resuspended in 50 μL 0.5 M EDTA (pH = 8.0) and heated for 5 min at 100 °C. The heated suspension was centrifuged at $12000 \times g$ for 10 min and the collected supernatant was incubated at 37 °C for 3 h with the addition of 10 μL 20 mg/mL proteinase K. After another incubation at 100 °C for 5 min, the PIA-containing supernatant was diluted by a factor of 40 and 400, respectively, and an aliquot of 5 μL was transferred to a nitrocellulose membrane (Millipore, HATF00010). After drying at 65 °C for 30 min, the membrane was incubated within 5% skim milk for 2 h at room temperature, and then washed three times with TBS. Then, 1 mL WGA-HRP conjugate solution (0.1 μg/mL in 5% skim milk) was added to the membrane surface and incubated for another 1 h with gentle shaking. After washing, electrochemiluminescence kit (Yeasen, Shanghai) was used for signal detection.

**Triton-X100-induced autolysis assay**. Overnight bacterial cultures were diluted into TSBg to a final optical density of 0.1 and the diluted cultures (3:2 flask/culture volume ratio) were then grown at 37 °C with gentle shaking (100 rpm). Cells from late exponential phase were harvested by centrifuging at $4000 \times g$ for 5 min and the cell pellet was washed three times with 100 mM Tris-HCl (pH 7.0) and then adjusted to $OD_{600} = 0.8$ by resuspending it in 100 mM Tris-HCl (pH 7.0) supplemented with 0.05% Triton-X-100. $OD_{600}$ was then measured at indicated intervals.

**Quantification of cell-free eDNA**. Quantification of cell-free eDNA was performed basically as described by DeFrancesco et al.[73], with some modifications. In brief, overnight grown bacterial cultures were diluted by 100-fold into TSB and then incubated at 37 °C, 200 rpm. Aliquots of culture were collected at indicated time and centrifuged at 4000 rpm for 5 min. Then, the bacterial pellet was resuspended in 0.9% NaCl and vortexed at the highest speed for 10 min. Then, the resuspension was filtered by 0.22-μm filter membrane. And 100 μL of the filtered resuspension was mixed with 100 μL of 1.5 μM PI. Fluorescence was measured by the synergy H1 plate reader (BioTeK, USA) with excitation and emission wavelengths of 535 and 617 nm, respectively. For absolute quantification of eDNA, denatured sperm DNA with a series of concentrations were used to generate the standard curve.

**Ethics approval**. This study was approved by the Ethics Committee of Renji Hospital, School of Medicine, Shanghai Jiao Tong University, Shanghai, China

(RA-2019-198). The bacterial isolates from patient samples were cultured and identified in routine microbiology laboratories. All animal work was approved by the ethics committee of Renji Hospital, School of Medicine, Shanghai Jiao Tong University, Shanghai, China.

**Statistics and reproducibility**. Results were shown as mean ± SD. Statistical analyses were performed using Student's $t$ test after Shapiro–Wilk normality test or two-way ANOVA with Bonferroni's multiple comparison post-test. Prism v7.0a was used to perform all statistical analysis. Sample size and type are both indicated in the corresponding figure legend.

**Reporting summary**. Further information on research design is available in the Nature Research Reporting Summary linked to this article.

## Data availability

Raw sequencing data are available on the Sequence Read Archive with the accession code PRJNA714433. Mass spectrometry-based omics data are available on the MetaboLights with the accession code MTBLS2979. Raw microscopy images are available on the figshare (https://doi.org/10.6084/m9.figshare.14821620). The source data are provided as Supplementary Data 1 and 2. Unedited blot and gel images are included as the Supplementary Fig. 4. All other data are available upon reasonable request from the corresponding author.

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

## Acknowledgements
This work was supported by the innovative research team of high-level local universities in Shanghai, the National Natural Science Foundation of China [grant numbers 81873957, 81861138043, 81802065, 81974311], the Shanghai Committee of Science and Technology, China [grant numbers 19JC1413005], and Shanghai Pujiang Program (2019PJD026).

## Author contributions
J.L. designed and performed the experiments and wrote the paper. Z.S., Q.H., Y.J., and Y. L. carried out the experiments and interpreted the data. Y.W., X.M., and Q.L. performed analyses of genomics and metabolomics data and provided technical assistance. J.T provided the catheter samples from hospitalized patients. L.H. and M.L. revised the manuscript and supervised the study.

## Competing interests
The authors declare no competing interests.
