## [Peer Review File · Communications Biology]

Reviewers' comments:

Reviewer #1 (Remarks to the Author):

General comments:

This study identifies a novel mechanism for small colony variant (SCV) formation in one of the most important culprits of implant-associated infections, *Staphylococcus epidermidis*. Genomic and metabolomics analyses reveal that a frame-shift mutation leading to impaired glycine production is the cause of SCV formation in this case. Accompanied with slow growth rate is also eDNA production caused by cell lysis, which leads to increased adhesion and biofilm formation. Ultimately, the biofilm production by SCV protects non-SCV during infections.

The study gives an important window into a new mechanism of SCV formation and how SCV leads to immune evasion of the faster-growing non-SCV. The study is scientifically sound and clearly communicated with a logical progression of research questions.

The study would benefit from a broader discussion about how cell lysis in the SCV strain might relate to metabolic shifts between lag-phase, exponential phase, and stationary phase to understand how the cell-specific growth rate affects its fate, i.e. whether it lyses or whether it lives. A couple of simple experiments can address whether cell lysis correlates with accumulation of specific metabolites, or whether it correlates with the growth rate.

Introduction:

Please specify which type of catheters (CV or urinary?).

Include background knowledge on stable and unstable SCV. What is known about the mechanisms for reverting into the NC phenotype and what affects the SCV's capability to do so? This background knowledge will allow the reader to assess if focusing on stable SCVs will focus the study on SCVs with particular mutations.

Results:

Line 180-182: "Acetyl-CoA was upregulated". Do you mean that Acetyl-CoA synthesis was upregulated? Which enzyme was upregulated? Perhaps you mean that the Acetyl-CoA concentration was higher in the mutant. If that is the case, please make this clear. It is confusing to say that a compound is up- or down-regulated. Similarly in line 188: "glycine was downregulated" – please write which enzyme and which process was down regulated.

Line 206 please spell Gram-positive with capital G. The method is named after Hans Christian Gram.

Figure 2: SCV were more adhesive and better at invasion. Were the physicochemical properties of the cell surface of SCV vs NC different from each other? If possible, please add characterization of the cell surface, e.g. water contact angle and zeta potential.

Figure 5a: Please include CLSM images with higher resolution, such that it is easier to determine the location of the DNA stained by PI. PI can sometimes cause intracellular staining, and it is also prone to some unspecific staining. It is difficult to evaluate any staining biases from the low-resolution images. The Syto9 stain indicates that there are no viable cells in the SCV biofilm. This observation should be discussed. Could it be possible that the "biofilm" is the remaining debris of an adhered population that has died?

If the amount of cells is similar in the two biofilms, it appears that the higher attachment rate (measured as initial attachment) does not correlate with more biofilm production.

Figure 5e: Please select different colors for the bars to make it easier to see which belongs to which sample.

Figure 6 (and throughout): When blue-red colors are used to distinguish between SCV and NC, please use the same color for each sample in all figures/sub-figures. I.e. always use red for SCV and blue for NC.

Figure 6H and G: Which is biofilm and which is autolysis? The figure caption says H for both.

Discussion:

It would be interesting to understand what the tipping-point is for whether cells survive or lyse. The authors discuss potential causes of cell lysis – but why do only some and not all of the cells in the population lyse? I suggest to consider the role of cell-specific growth rates in answering this question. Growth rates only declined in the absence of glycine, and the authors explain this by a lower peptidoglycan synthesis rate. However, it is also possible that cells that had faster peptidoglycan synthesis lysed because they could not form the penta-glycine crosslink at the same speed. If so, only the slow-growing population would survive and would soon represent the main population. In my opinion, it should not be assumed that the entire population has shifted to a slower growth rate. Consider figure 4D, which shows the distribution of peptidoglycan synthesis in single cells. NC also has slow-growing cells, but in much less proportion of the total population. If fast-growing cells die in the SCV, one would expect to see the distribution shift as is depicted.

If SCV lyse when growing too fast, the growth conditions will have a big influence on how many of the SCV population undergo cell lysis during the incubation. Cell lysis would peak in the exponential growth phase, which could be quantified by measuring the concentration of extracellular DNA at different time points during batch growth. Please perform this experiment.

Furthermore, if cell lysis is linked to growth rate, it is also possible that glucose (or acetate) addition to the growth media leads to increased cell lysis – not due to accumulation of acetic acid, but due to the imbalance in peptidoglycan synthesis which becomes fatal at high growth rates.

What is the expected variation in CidA transcription during log, exponential and stationary growth phases? Is CidA involved in peptidoglycan remodeling required for cell growth? Can changes in CidA production simply reflect growth rate?

Acetate accumulates during exponential growth of *S. aureus*. Is this also the case for *S. epidermidis*? If so – the correlation between acetate production and cell lysis is inevitable. Again, analysis of the timing of cell lysis vs the accumulation of acetate during growth in a batch culture will reveal if the two are truly correlated, or if cell lysis commences in the exponential phase prior to accumulation of acetate.

Metabolic imbalances in general can also cause accumulation of reactive oxygen species (ROS). Did you measure ROS?

Methods:

What were the criteria for validating colony diameter measurements? If colonies are close to each other (such as in Figure 4B) they will be smaller. Therefore, colony diameter must be measured for colonies that are well separated from other colonies to avoid this bias. Please clarify if/how you avoided this bias in the data.

Line 553: Biofilms were grown by inoculating cultures after adjusting to OD 0.1. If the SCV culture already contained much eDNA, this would also be present in the inoculum.

Line 556: What is the fixative agent used, and does it affect viability staining?

Line 558: What was crystal violet dissolved in? how was crystal violet extracted before quantification? Were biofilms washed before extracting the stain? Was the extracted stain measured in a new tray, or did the optical density represent absorbance from the stain + the bacteria?

Line 590 onwards: Primary attachment is measured after diluting overnight cultures with the same dilution factor, followed by quantifying the number of attached cells by CFU enumeration. Why did you not use confocal microscopy to visualize the attached cells? If SCV not only grow more slowly but are at risk of death by lysis, the lower CFU count would reflect a combination of adhesion + survival, which may not be identical for the two strains. Direct enumeration by microscopy would give a better analysis of bacterial attachment.

For each statistical analysis done, please write what the number of replicates was. Please clarify if they are technical replicates (several samples from e.g. same bacterial culture) or whether they are biological replicates (experiments performed with individually grown cultures). If T-tests are performed, please describe which analysis was done to determine that the data was normally distributed.

Reviewer #3 (Remarks to the Author):

The manuscript of Liu et al describes a deep characterization of one clinical stable SCV in comparison to the normal counterpart NC. The authors investigated the metabolic pathways involved in each species and found out that the stable SCV is auxotroph for glycine biosynthesis. This mutation is responsible for slow growth because affects the peptidoglycan-linking rate. On the other hand, stable SCV presents a high production of pyruvic acid and acetyl-CoA which leads to high production of acetate under biofilm conditions, a main bacterial characteristic associated to catheter infections. The authors found that the stable SCV contributes to biofilm formation due to eDNA release produced by autolysis.

The manuscript is well written and easy to follow. However, I have some comments and suggestions:
Results:

Table 1: why the authors used different test to analyze the differences?

Fig. 1/mat and methods: the stability test is not well described. How many times did the authors streak the SCVs to investigate the stability? Is it done for 24h or more subcultivation steps were done?

Line 118: please add a reference for SCV characteristics

Line 124: Should be Table S1

The supplementary figures are missing (figures S1, 2)

Line 158: is Fig S2 or Table S2?

How was calculated the invasion rate? Which MOI was used in these experiments? I assume that both strains grow with different dynamic, are the cells infected with similar amount of CFU?

Fig. 3B: please remove the lines around the figure

Fig. 3D: Are the Nc1 to 4 replicates?

Line 178: is it correct to add here fig.3D or E?

Line 182: do you refer to fig. 3F?

Did the authors check the glycine auxotrophism in other clinical SCVs? The authors should notice that the lack of glycine production in SCVs was described by Lannergard et al 2008 in *S. aureus* (doi: 10.1128/AAC.00668-08)

Please add which type of statistic test was done in the figures 4 and 5

Fig. 4: Please check that the format of fig. 4A. Why the pictures from the colonies of SCV on TSA look much bigger than in figure 1?

Line 246: add after biofilm formation (Fig. 2D) to guide the reader

Line 245-247: From which experiments the authors found that the enhanced biofilm resulted from increased release of extracellular matrix and not from the number of cells?

Fig. 5E: please include the controls without treatment to help the reader for the interpretation of this figure

Fig. 6E: did the authors analyzed whether those curves are significantly different by statistical analysis?

Fig. 6F: how was the comparison? Can the authors add controls without treatment? Please check that the figures 6B and F represent the data in similar order to avoid confusions

The legend of the figure 6H and G are not correct

Discussion:

Line 383: please add a reference (catheter infection in hyperglycemic patients)

Line 391: is it possible to include the data not shown in suppl material?

Line 431: is the word "lies" appropriated here?

Line 440-444: the sentence that starts with "In general..." is confused. Please re-write

Line 462: please add a reference about the role of glycine in cell wall biosynthesis

Ethic approval: please include in this section the ethic approval for animal's experiments. Please add the number of the study approval as well as animal approval.

Reviewer #1 (Remarks to the Author):

General comments:

This study identifies a novel mechanism for small colony variant (SCV) formation in one of the most important culprits of implant-associated infections, *Staphylococcus epidermidis*. Genomic and metabolomics analyses reveal that a frame-shift mutation leading to impaired glycine production is the cause of SCV formation in this case. Accompanied with slow growth rate is also eDNA production caused by cell lysis, which leads to increased adhesion and biofilm formation. Ultimately, the biofilm production by SCV protects non-SCV during infections.

The study gives an important window into a new mechanism of SCV formation and how SCV leads to immune evasion of the faster-growing non-SCV. The study is scientifically sound and clearly communicated with a logical progression of research questions.

The study would benefit from a broader discussion about how cell lysis in the SCV strain might relate to metabolic shifts between lag-phase, exponential phase, and stationary phase to understand how the cell-specific growth rate affects its fate, i.e. whether it lyses or whether it lives. A couple of simple experiments can address whether cell lysis correlates with accumulation of specific metabolites, or whether it correlates with the growth rate.

We thank the reviewer for all the constructive comments and suggestions to improve our manuscript. We have attempted to address the questions either in the main body of the manuscript, or in the point-by-point response letter. We hope that our revisions and explanations adequately respond to the comments and enhance the manuscript.

Introduction:

Please specify which type of catheters (CV or urinary?).

Response: We thank the reviewer for this comment. The *S. epidermidis* SCV strain was isolated from central venous catheter (CVC), and now we specify this point in the “Introduction” section (Line101).

Include background knowledge on stable and unstable SCV. What is known about the mechanisms for reverting into the NC phenotype and what affects the SCV's capability to do so? This background knowledge will allow the reader to assess if focusing on stable SCVs will focus the study on SCVs with particular mutations.

- Response: We thank the reviewer for the insightful comment. Now we have concisely introduced unstable SCV and stable SCV (Line71-80). The introduction goes like:

“Although these stable SCVs mutants have provided profound insights into lifestyle of SCVs, of particular note is that mutations associated with SCVs formation are usually unstable. As such, SCVs can easily revert to normal colony phenotype via reversal of mutations or gain of second-site mutations . In other words, most SCVs are phenotypically unstable after serial passages. Obviously, such instability hampers in-depth characterization of SCVs, thereby leading to the fact that numerous studies established their conclusions based on phenotypically stable ETC mutants.”

Results:

Line 180-182: “Acetyl-CoA was upregulated”. Do you mean that Acetyl-CoA synthesis was upregulated? Which enzyme was upregulated? Perhaps you mean that the Acetyl-CoA concentration was higher in the mutant. If that is the case, please make this clear. It is confusing to say that a compound is up-or down-regulated. Similarly in line 188: “glycine was downregulated” – please write which enzyme and which process was down regulated.

Response: We thank the reviewer and apologize for the perplexing word usage. Actually, we intended to mean that it is the concentration of Acetyl-CoA that increase. Now we add “the concentration of” in the corresponding passage.

Line 206 please spell Gram-positive with capital G. The method is named after Hans Christian Gram.

Response: We apologize for the mistake. We have replaced 'gram' with 'Gram'.

Figure 2: SCV were more adhesive and better at invasion. Was the physicochemical properties of the cell surface of SCV vs NC different from each other? If possible, please add characterization of the cell surface, e.g. water contact angle and zeta potential.

Response: We thank the reviewer for this insightful comment. It is likely that surface characteristics are different between them, since bacterial attachment to surface is a process quite affected by surface properties. We have investigated zeta potential and the result (as shown below) demonstrates that the stable SCV was more positive than NC, with a potential of -5.83 mV and -15.9 mV, respectively. Considering that most kinds of material was negative (< -23.61 mV)¹, the repulsive force may be reduced between the more positively charged SCV cell and the substratum. We have added this characterization in the manuscript (Line 155-161) and the zeta potential distribution is now supplemented as figure S1B and S1C.

Zeta potential distribution of SCV and NC. Mean values of zeta potential were indicated.

(FigureS1B, C in the resubmission)

1. Chia TW, Nguyen VT, McMeekin T, Fegan N, Dykes GA. Stochasticity of bacterial attachment and its predictability by the extended derjaguin-landau-verwey-overbeek theory. *Appl Environ Microbiol* 77, 2011, 3757-64.

Figure 5a: Please include CLSM images with higher resolution, such that it is easier to determine the location of the DNA stained by PI. PI can sometimes cause intracellular staining, and it is also prone to some unspecific staining. It is difficult to evaluate any staining biases from the low-resolution images. The Syto9 stain indicates that there are no viable cells in the SCV biofilm. This observation should be discussed. Could it be possible that the “biofilm” is the remaining debris of an adhered population that has died?

Response: We thank the reviewer for this comment. We have tried our best to improve the resolution of CLSM images and now it should have higher resolution in the new Figure5. We agree with reviewer that PI can cause staining biases, but observational findings from CLSM images, such as few alterations in PIA level, much lower live/dead cells ratio and higher eDNA concentration were all verified by further experiments. Particularly, Syto9-stained live cells were quite few but not none. CLSM images from another replicate shows that live cells yet remain, but still at a lower number than that in NC-forming biofilm. Now CLSM images from the more representative replicate has been included in the new Figure5. Debris from dead cells consists of a large part of SCV-forming biofilm and live cells are quite few there, therefore, to some extent, biofilm of SCV could only be referred to as “film” since non-biological entity represents the most part of it. However, if we consider SCV as a whole with NC, it should serve as an integral part of biofilm.

If the amount of cells is similar in the two biofilms, it appears that the higher attachment rate (measured as initial attachment) does not correlate with more biofilm production.

Response: We thank the reviewer for raising this question. It is true that more biofilm production cannot be solely attributable to the higher attachment rate from our existing data. But in consideration of the growth defect of the stable SCV, cell quantity would be lower in SCV biofilm if the two strains had the same ability to establish and maintain biofilm, therefore SCV should have powerful ability to elaborate biofilm, such as higher attachment ratio. Nevertheless, eDNA is considered as the multifaceted matrix to not

only aid attachment but also secure structural stability, thus we believe that eDNA can facilitate SCV biofilm formation by influencing a multitude of other processes in addition to primary attachment.

Figure 5e: Please select different colors for the bars to make it easier to see which belongs to which sample.

Response: Thanks for the good advice. We have updated Figure5e and now it may be easily discernible to differentiate between the sample types.

Figure 6 (and throughout): When blue-red colors are used to distinguish between SCV and NC, please use the same color for each sample in all figures/sub-figures. I.e. always use red for SCV and blue for NC.

Response: We thank the reviewer for their comments and apologize for the confusion. Colors of all figures are carefully checked and updated.

Figure 6H and G: Which is biofilm and which is autolysis? The figure caption says H for both.

Response: We thank the reviewer for the comment and apologize for not correctly indicating the figure captions. They have been corrected in the resubmission.

Discussion:

It would be interesting to understand what the tipping-point is for whether cells survive or lyse. The authors discuss potential causes of cell lysis – but why do only some and not all of the cells in the population lyse? I suggest to consider the role of cell-specific growth rates in answering this question. Growth rates only declined in the absence of glycine, and the authors explain this by a lower peptidoglycan synthesis rate. However, it is also possible that cells that had faster peptidoglycan synthesis lysed because they could not form the penta-glycine crosslink at the same speed. If so, only the slow-

growing population would survive and would soon represent the main population. In my opinion, it should not be assumed that the entire population has shifted to a slower growth rate. Consider figure 4D, which shows the distribution of peptidoglycan synthesis in single cells. NC also has slow-growing cells, but in much less proportion of the total population. If fast-growing cells die in the SCV, one would expect to see the distribution shift as is depicted.

If SCV lyse when growing too fast, the growth conditions will have a big influence on how many of the SCV population undergo cell lysis during the incubation. Cell lysis would peak in the exponential growth phase, which could be quantified by measuring the concentration of extracellular DNA at different time points during batch growth. Please perform this experiment.

Furthermore, if cell lysis is linked to growth rate, it is also possible that glucose (or acetate) addition to the growth media leads to increased cell lysis – not due to accumulation of acetic acid, but due to the imbalance in peptidoglycan synthesis which becomes fatal at high growth rates.

Response: We thank the reviewer for raising such a valid notion. It would be of great interest if autolysis and subsequent eDNA release of SCV were attributable to phase-specific growth rates, which is closely associated with the deficiency in peptidoglycan synthesis.

Now we have performed additional experiments, such as measurement of cell-free eDNA in culture medium, to ascertain the autolysis dynamic. From the dynamic curve of cell-free eDNA concentration (Figure6B in the resubmission) (as shown in Figure1 below), we can clearly notice that eDNA concentration is nearly consistent between SCV and NC during lag phase, but SCV begin to quickly autolyze at the early exponential growth phase and the concentration of eDNA is continuously increasing during all later growth phases, even when entering into stationary growth stage, which indicates autolysis of SCV constantly exists, and always at a higher speed than that of NC.

Concentration of cell-free eDNA from batch cultures at indicated time points (Figure6B in the resubmission).

To confirm such result, flow cytometry analysis of cell viability, which is more sensitive to the event of cell autolysis, was also performed. As shown below, Dead population of SCV is constantly increasing during the late exponential phase (from 6h to 12h) and continue increasing even at stationary stage (24h), whereas NC almost do not autolyze until stationary stage. Despite that, there is a slight reduction in stationary autolysis rate of SCV (Dead population increase by 3.07% in 12 hours), as compared with that of exponentially growing SCV (Dead population increase by 4.39% in 6 hours).

Cell viability analysis by flow cytometry. Percent of dead population is indicated at corresponding location (FigureS3B in the resubmission).

Results from the two supplementary experiments can lead us to the conclusion that autolysis of SCV is constantly happening after lag phase, but autolysis of SCV at stationary growth stage is not as quick as that of exponentially growing SCV, which may be attributable to the phase-specific growth rate (the higher, the quicker). It is also likely that excess glucose in the growth media would increase cell lysis by accelerating growth, but rapid autolysis of SCV cannot be solely attributed to specific growth rates, since autolysis at stationary growth phase still exists, thus another cause, such as accumulation of acetic acid, may still play a role.

What is the expected variation in CidA transcription during log, exponential and stationary growth phases? Is CidA involved in peptidoglycan remodeling required for cell growth? Can changes in CidA production simply reflect growth rate?

Response: CidA, which exhibit holing-like properties, is known to be involved in *S. aureus* biofilm formation by controlling cell lysis and the release of genomic DNA. CidA can oligomerize into high-molecular-mass complexes and induce cell death and lysis in a way similar to bacteriophage.

But actually, functional roles that CidA play in *S. epidermidis* is not well understood. Based on the existing characterization of *S. aureus* CidA, there is no definitive link between CidA and peptidoglycan remodeling. Transcription of *S. aureus* CidA would peak at the exponential growth phase and remain low at both lag and stationary growth phases, thus production of CidA can, to some extent, simply reflect the growth rate,

Acetate accumulates during exponential growth of *S. aureus*. Is this also the case for *S. epidermidis*? If so – the correlation between acetate production and cell lysis is inevitable. Again, analysis of the timing of cell lysis vs the accumulation of acetate during growth in a batch culture will reveal if the two are truly correlated, or if cell lysis commences in the exponential phase prior to accumulation of acetate.

Response: In our case of *S. epidermidis*, concentration of extracellular acetate also peaks at exponential growth phase. Specifically, acetate begin to accumulate during lag phase,

and peak at later exponential growth phase. Thus the correlation between acetic overproduction and cell lysis cannot be completely excluded since SCV begin to autolyze at the early exponential growth stage.

Metabolic imbalances in general can also cause accumulation of reactive oxygen species (ROS). Did you measure ROS?

Response: We greatly appreciate the reviewer for the reasonable comment. Reactive oxygen species is a quite powerful indicator for cellular metabolic state and its excess is cyto-toxic. Thus we have now determined endogenous ROS level of overnight grown NC and SCV using the cell permeant reagent 2',7'-dichlorofluorescein diacetate. As shown below, intracellular ROS level of SCV is higher than that of NC, which is indicated by the higher fluorescence intensity from the FITC channel. This result further confirmed the metabolically dysfunctional state of SCV and we add it as the supplementary figure S2B in the revision.

Intracellular ROS level (Figure S2B in the resubmission)

Methods:

What were the criteria for validating colony diameter measurements? If colonies are close to each other (such as in Figure 4B) they will be smaller. Therefore, colony diameter must be measured for colonies that are well separated from other colonies to avoid this bias. Please clarify if/how you avoided this bias in the data.

Response: We thank the reviewer for the thoughtful comment. We agree with reviewer that colonies from overcrowded area would introduce measuring bias. To minimize such potential bias, we have re-performed the experiment by plating serially diluted cultures onto agar plates to ensure all plates to generate similar CFU. However, it is not yet completely free of subjective bias when colony diameters were quantified using ImageJ, as overcrowded regions still remain on the agar plates and it is still hard to objectively define the “well-separated colonies”. Considering that alterations in growth dynamics can be well reflected by the more objectively quantitative growth curve, we have removed data from colony diameter measurements and only presented snapshots of agar plates as the supplementary data in the resubmission (FigureS2C).

Line 553: Biofilms were grown by inoculating cultures after adjusting to OD 0.1. If the SCV culture already contained much eDNA, this would also be present in the inoculum.

Response: It is true that the initial amount of eDNA presented in the inoculum of SCV is high, but the concentration difference of eDNA between the two strains at the beginning is not as striking as that in later phase, which can demonstrate that SCV has stronger ability to generate much eDNA during the course of growth than NC. Particularly, the inoculum was from overnight grown culture, from which eDNA concentration was not as high as that from culture grown for 24h.

Line 556: What is the fixative agent used, and does it affect viability staining?

Response: Bouin’s fixative was used as the fixative agent and it is a mixture of formaldehyde, picric acid and acetic acid. The formula is as below:

Saturated picric acid solution	75mL
Formaldehyde (3.7-4%)	25mL

Acetic acid ($\geq 99.5\%$)	5 mL
Total volume	105mL

Like other types of fixative, Bouin's fixative potentially interfere with cell viability as well. It is also the reason why we did not use Bouin's fixative when observing live/dead cells within biofilm by CLSM. However, for crystal violet-assisted semi-quantification of biofilms, Bouin's fixative is commonly used, because it is able to preserve cell morphology and prevent cell from autolysis, thereby ensuring future analysis after biofilm has been stained.

Line 558: What was crystal violet dissolved in? how was crystal violet extracted before quantification? Were biofilms washed before extracting the stain? Was the extracted stain measured in a new tray, or did the optical density represent absorbance from the stain + the bacteria?

Response: Crystal violet was dissolved in 1% ammonium oxalate solution containing 20% absolute ethanol. After crystal violet completed staining biofilm, biofilms were washed a couple of times to remove residual crystal violet, thus the optical density represents absorbance from the crystal violet-positive bacteria and biofilm matrix.

To make the description about crystal violet staining much clear, now it goes like:

The fixative was gently aspirated out and wells were washed three times with PBS, and stained with 0.4% (wt/vol) crystal violet dissolved in 1% ammonium oxalate solution containing 20% absolute ethanol. Biofilm formation was measured by a MicroELISA autoreader (BioTeK, USA) at 570 nm after being washed by PBS to remove redundant crystal violet.

Line 590 onwards: Primary attachment is measured after diluting overnight cultures with the same dilution factor, followed by quantifying the number of attached cells by CFU enumeration. Why did you not use confocal microscopy to visualize the attached cells? If SCV not only grow more slowly but are at risk of death by lysis, the lower CFU

count would reflect a combination of adhesion + survival, which may not be identical for the two strains. Direct enumeration by microscopy would give a better analysis of bacterial attachment.

Response: We completely agree with the reviewer that survival ability would interfere with the result. In order to minimize the interference, in fact, we shortened incubation time for attachment (only one hour). By doing so, quite low CFU was obtained from each well after washing and this would result in difficulty in searching for good microscope view and numerating enough cells. Further, if SCV had identical adhesive ability to NC but quicker lysis than NC, lower attachment rate would be obtained. Considering the experimentally measured higher attachment rate of SCV, its actual attachment rate should be high enough to counteract effect from autolysis. It is true that microscopy-assisted enumeration can generate a better analysis, yet CFU enumeration alone can potentially prove SCV's higher attachment ability.

For each statistical analysis done, please write what the number of replicates was. Please clarify if they are technical replicates (several samples from e.g. same bacterial culture) or whether they are biological replicates (experiments performed with individually grown cultures). If T-tests are performed, please describe which analysis was done to determine that the data was normally distributed.

Response: We thank the reviewer for the thoughtful suggestion. Now both type and number of replicates are indicated at the Figure Legend section. Statistical analysis for normality test is also specified.

Reviewer #3 (Remarks to the Author):

The manuscript of Liu et al describes a deep characterization of one clinical stable SCV in comparison to the normal counterpart NC. The authors investigated the metabolic pathways involved in each species and found out that the stable SCV is auxotroph for glycine biosynthesis. This mutation is responsible for slow growth because affects the peptidoglycan-linking rate. On the other hand, stable SCV presents a high production of pyruvic acid and acetyl-CoA which leads to high production of acetate under biofilm conditions, a main bacterial characteristic associated to catheter infections. The authors found that the stable SCV contributes to biofilm formation due to eDNA release produced by autolysis.

The manuscript is well written and easy to follow. However, I have some comments and suggestions:

We would like to thank the reviewer for all the kind comments and helpful suggestions supporting our manuscript. Following the comments, we have revised the manuscript accordingly and given explanations in this point-by-point reply letter. We hope that these revisions can address the reviewer's concerns.

Results:

Table 1: why the authors used different test to analyze the differences?

Response: We thank the reviewer for the comment. Indeed, we performed different statistical test because chi-square test is applicable to comparison of the frequencies of various categories (e.g. Female/Male, Yes/No), whereas one-way ANOVA is suitable for comparison of the differences of measurement data (e.g. Age) between three groups. The given P value would be more accurate if suitable statistical tests were chosen according to the data type.

Fig. 1/mat and methods: the stability test is not well described. How many times did the authors streak the SCVs to investigate the stability? Is it done for 24h or more subcultivation steps were done?

Response: We thank the reviewer for the careful consideration. At the initial stage of stability-checking experiment, SCVs were only streaked to another sheep blood agar plate at a time and incubated for 24h. Thus it is done for 24h and not any other sub-cultivation steps were done. By this way, several SCV cases have been selected, including the SCV case deeply characterized in this study.

Although stability was only checked once at the beginning of the study, the deeply investigated SCV case was later found to be phenotypically stable even after three times of sub-cultivation.

To make it more clear, the description of stability test (Line) now goes like:

“at least 50 SCVs from the specimen were streaked onto another 5% sheep blood agar plate, then incubated for 24 hours, and simultaneously each plate was streaked with a homologous normal colony to facilitate further stability analysis of SCVs.”

Line 118: please add a reference for SCV characteristics

Response: We thank the reviewer for the thoughtful suggestion. Now four related references were added in the corresponding passage (Line 135 in the resubmission).

Line 124: Should be Table S1

Response: We feel sorry for the mistake. Now it has been corrected.

The supplementary figures are missing (figures S1, 2)

Response: We apologize for the oversight. We missed uploading image files of Figures S1 and S2. Now they are both uploaded in the revision.

Line 158: is Fig S2 or Table S2?

Response: We feel sorry for this confusion. It should be Table S2 there. Now it has been corrected.

How was calculated the invasion rate? Which MOI was used in these experiments?

Response: We thank the reviewer for the comment. The invasion rate was determined as the percentage of CFU counts of intracellular bacteria among CFU counts of bacteria that were initially added into cells. And MOI of 100 bacteria per cell was used in these experiments. Descriptions about both calculation of the invasion rate and MOI value are now supplemented in the “materials and methods” section (Line 679 in the resubmission).

I assume that both strains grow with different dynamic, are the cells infected with similar amount of CFU?

Response: We agree with the reviewer that both strains have different growth dynamics. However, optical density of exponential cultures of both strains had been adjusted to the same value ($OD_{600} = 0.4$) before incubation with cells started and simultaneously, appropriate dilutions of the OD_{600} -adjusted bacterial cultures were also prepared for CFU counting and CFUs of the bacterial inoculum were counted after incubation and found to be similar between both strains, thus both strains shared similar amount of CFU to infect cells.

Fig. 3B: please remove the lines around the figure

Response: We thank the reviewer for the advice. Now the lines around the figure were completely removed.

Fig. 3D: Are the Nc1 to 4 replicates?

Response: It is true that the number after “NC” indicated different replicates. We now specify this point the figure 3E legend.

Line 178: is it correct to add here fig.3D or E?

Response: We agree with the reviewer that fig.3E should also be included. Now “and Figure 3E” is inserted in the resubmission.

Line 182: do you refer to fig. 3F?

Response: Actually, exact numbers of fold change of differentially expressed metabolites were not given in figures/tables of this manuscript, but Log₂ fold change can be estimated from the color bar of Figure 3E, where acetyl-CoA holds most red color (i.e. the highest positive fold change)

Did the authors check the glycine auxotrophism in other clinical SCVs? The authors should notice that the lack of glycine production in SCVs was described by Lannergard et al 2008 in S. aureus (doi: 10.1128/AAC.00668-08)

Response: We thank the reviewer for the insightful comment. We have checked other clinical SCVs that we collected to ascertain whether other SCVs have glycine-auxotrophism or not. Unfortunately, there is only one another glycine-auxotrophic *S. epidermidis* SCV case, but we did not perform whole-genome sequencing for this SCV strain since this SCV strain does not have other interesting phenotype, such as increased biofilm-forming ability. Thus we did not discuss too much about this strain due to the information shortage of genetic background.

Please add which type of statistic test was done in the figures 4 and 5

Response: We appreciate the reviewer's comment. Unpaired Student's t test was used for these statistical analyses. We now supplemented the statement in the Figure legend.

Fig. 4: Please check that the format of fig. 4A. Why the pictures from the colonies of SCV on TSA look much bigger than in figure 1?

Response: We apologize for the mistake. There might be something wrong when assembling and exporting images of figure1 and figure4. Now we have arranged, assembled and exported both Figure1 and Figure4 (now FigureS2C in the resubmission) under the same condition. Now the inaccuracy should be corrected in the revision.

Line 246: add after biofilm formation (Fig. 2D) to guide the reader

Response: We thank the reviewer for the comment. We have added it after "biofilm formation"

Line 245-247: From which experiments the authors found that the enhanced biofilm resulted from increased release of extracellular matrix and not from the number of cells?

Response: We apologize for making this misleading statement. We would like to clarify that at Line 245-247, it is only the hypothesis and more verifiable experiments were described at later passages. Considering that SCV grows slowly, it would generate much lower biofilm content if it had the identical matrix composition or other biofilm properties to NC. Yet SCV has much higher biofilm content, this indicates that SCV may have special matrix composition or variation of other properties as compared with NC. Now this statement has been corrected as below:

“The observed growth defect of the *S. epidermidis* stable SCV isolate led us to propose that its enhanced biofilm formation (Figure 2A) resulted from increased release of extracellular matrix instead of an increase in the number of cells. “

Fig. 5E: please include the controls without treatment to help the reader for the interpretation of this figure

Response: We have added data from the group without any treatment as the control.

Fig. 6E: did the authors analyzed whether those curves are significantly different by statistical analysis?

Response: We thank for the comment. We have performed statistical analyzes by using two-way ANOVA with Tukey's multiple comparison post-test and p values are indicated in the new figure 6E (now Figure 6F)

Gig. 6F: how was the comparison? Can the authors add controls without treatment?

Response: We thank the reviewer for this comment. The comparison was made by comparing fold change with no treatment group. Now we have add the group without any treatment as the control.

Please check that the figures 6B and F represent the data in similar order to avoid confusions

Response: We thank for the advice. We have carefully checked data from the two figures and found that there is nothing wrong.

The legend of the figure 6H and G are not correct

Response: We apologize for creating this confusion. We have corrected the typing error in the resubmission.

Discussion:

Line 383: please add a reference (catheter infection in hyperglycemic patients)

Response: We thank the reviewer for the constructive suggestion. Now an item of related reference was added:

Marsillio, L. E. et al. Hyperglycemia at the Time of Acquiring Central Catheter-Associated Bloodstream Infections Is Associated With Mortality in Critically Ill Children. *Pediatr Crit Care Med* 16, 621-628, doi:10.1097/pcc.0000000000000445 (2015)

Line 391: is it possible to include the data not shown in suppl material?

Response: We thank the reviewer for the comment. We feel that it is somewhat inappropriate to publish the data, since there are only data from one biological replicate for both cases. Nevertheless, it is expectable that detailed characterization of more SCV cases would be included in our future work.

Line 431: is the word “lies” appropriated here?

Response: We agree with the reviewer that “lies” was inappropriately used there. Now it has been replaced by “is”.

Line 440-444: the sentence that starts with “In general...” is confused. Please re-write

Response: We apologize for creating this confusion. “In general” is redundant in the corresponding sentence and now we delete it.

Line 462: please add a reference about the role of glycine in cell wall biosynthesis

Response: We thank the reviewer for the good suggestion. Now an item of related reference was added:

Hammes, W., Schleifer, K. H. & Kandler, O. Mode of action of glycine on the biosynthesis of peptidoglycan. *J Bacteriol* 116, 1029-1053, doi:10.1128/jb.116.2.1029-1053.1973 (1973)

Ethic approval: please include in this section the ethic approval for animal's experiments. Please add the number of the study approval as well as animal approval.

Response: We thank the reviewer for this comment. We have added the number of the study approval (RA-2019-198) in the 'Ethics approval' section. However, unlike human-associated study, the number of animal study approval has not been provided by the Ethics Committee. Now the statement that all animal work was approved by the ethics committee has been supplemented to the 'Ethics approval' section.

REVIEWERS' COMMENTS:

Reviewer #1 (Remarks to the Author):

The authors have carefully addressed all comments to the manuscript and followed up with additional experiments as suggested. I think the revised manuscript has further strengthened the discussion and supports the conclusions drawn by the authors.

Reviewer #3 (Remarks to the Author):

The authors answered all my questions and improved the manuscript. I recommend to accept the manuscript in its current form